# Accelerating cell culture media development using Bayesian optimization-based iterative experimental design

Harini Narayanan [1], Joshua A. Hinckley[1,2], Rachel Barry[1,2], Brendan Dang[2], Lenna A. Wolffe [2], Adel Atari[2], Yuen-Yi Tseng[2] & J. Christopher Love [1,3] ✉

Optimizing operational conditions for complex biological systems used in life sciences research and biotechnology is an arduous task. Here, we apply a Bayesian Optimization-based iterative framework for experimental design to accelerate cell culture media development for two applications. First, we show that this approach yields new compositions of media with cytokine supplementation to maintain the viability and distribution of human peripheral blood mononuclear cells in the culture. Second, we apply this framework to optimize the production of three recombinant proteins in cultivations of *K.phaffii*. We identified conditions with improved outcomes for both applications compared to the initial standard media using 3–30 times fewer experiments than that estimated for other methods such as the standard Design of Experiments. Subsequently, we also demonstrated the extensibility of our approach to efficiently account for additional design factors through transfer learning. These examples demonstrate how coupling data collection, modeling, and optimization in this iterative paradigm, while using an exploration-exploitation trade-off in each iteration, can reduce the time and resources for complex optimization tasks such as the one demonstrated here.

Cell culture is an essential technique used throughout life sciences to study cellular, molecular, and disease biology. It is also a critical unit operation in biotechnology used to manufacture a wide range of products such as therapeutics, food proteins, peptides, biofuels, metabolites, industrial enzymes, biomass, and cells themselves for cell therapy, and the artificial meat industry[1,2]. The medium provides the essential nutrients and elements required for the ex-vivo growth and proliferation of the cells, the production of intended compounds, and the quality of the product[3,4].

Optimizing the compositions of these media is a common challenge across applications. The components required are manifold, including nutrients such as amino acids, nitrogen sources, carbon sources, minerals, salts, and growth hormones, among many others. This diversity leads to a large number of design factors ranging from about 10's-100's media components, presenting a highly combinatorial design space with complex interactions[2]. Additionally, selecting the most suitable media significantly depends on the type and lineage of the cells, the specific objective of the cell culture (such as homeostasis, growth, or differentiation), and the required operating conditions. These factors together render this optimization task a resource-intensive and laborious one. To address this challenge, standard practices across fields rely on either the use of well-documented, historical formulations of 'universal' standard media[3] or formulations resulting from a limited optimization using one factor at a time (OFAT), or a statistical Design of Experiments (DoEs)[5-9].

Algorithms for metaheuristic optimization have been used as an alternative to these conventional approaches[3,10], particularly for cases of bacterial cultivation and fermentation applications[2,3]. These

[1]Koch Institute for Integrative Cancer Research at MIT, 500 Main Street, Cambridge, MA 02139, USA. [2]Broad Institute of MIT and Harvard, 415 Main Street, Cambridge, MA 02142, USA. [3]Department of Chemical Engineering, Massachusetts Institute of Technology, Cambridge, MA 02139, USA. ✉e-mail: clove@mit.edu

algorithms are often combined with surrogate models, such as quadratic response surface methodology (RSM), Artificial Neural Networks (ANNs), or tree-based approaches[11], to represent the underlying relationship between design factors and the target objective. These studies, however, decouple data collection, modeling, and optimization. Data are first collected using one of the statistical DoE approaches and subsequently used for model development followed by optimization focused on maximizing or minimizing a desired target[2,3,11]. This staging requires significant coverage of the design space with the collected data to build robust surrogate models and avoid local regions of suboptimal solutions. Furthermore, these approaches have a limited ability to representatively account for the intrinsic noise in biological datasets when developing the surrogate model and performing the optimization[3]. Finally, all the methodologies, including OFAT and DoE, use only two types of design factors (continuous and discrete)[3,12]. Certain media components present multiple formats from which to choose, such as the type or source of carbon (e.g., glucose, glycerol, lactate, fructose, etc.) or nitrogen (e.g., ammonium salts, urea, glutamine). This representation of identity introduces categorical design factors that OFAT and DoE are not designed to accommodate[3,12], and modification to these approaches to account for categorical factors scales the design space quickly with an increasing number of categories and multiple categorical variables. Even considering continuous variables, these approaches suffer from a larger number of factors (>15–20)[13] resulting in approximations based on preliminary screening designs[14]. The inherent linear/quadratic response surface assumptions of the DoEs coupled with these additional biases introduced to make the design space feasible results in DoEs being in-efficient and suboptimal. Furthermore, these approaches are not designed to plan experiments for constrained design spaces[12,15] and have limited capabilities to accumulate and propagate knowledge, emphasizing the need for alternative resource-efficient experimental design approaches for optimization.

To address these challenges, we demonstrate here an iterative approach to experimental design that relies on Bayesian Optimization (BO)[16]. This strategy provides two key benefits. First, the use of a probabilistic surrogate model (Gaussian Process (GPs)[17]) that is particularly well-suited for biological applications. GPs are suitable for unbiased learning of smooth response functions compared to alternative ML algorithms, such as tree-based models that are bound by splitting rules and learn discontinuous or piecewise continuous decision boundaries. Furthermore, GPs can include prior beliefs about the system, incorporate process noise in its implementation, and obtain confidence in its predictions by associating higher uncertainty with unexplored parts of the design space[18]. Most other classical ML models don't allow the explicit incorporation of prior assumptions or process noise and inherently provide point estimates[19]. While uncertainty can be estimated using ensembling techniques, these estimates do not directly correspond to the positions of the data points within the design space. These abilities are important for intrinsically noisy biological systems that require expensive experimentation. In this context, approaches that can encode prior beliefs could reduce the overall experimental burden of optimization. Furthermore, GPs are efficient for handling small volumes of data (common with biological systems) compared to alternative tree-based approaches, which often perform well with larger volumes of data (e.g. a pre-existing database is available)[17]. Additionally, custom kernels can be designed for GP models to suit the specific needs of an application.

Second, while planning new experiments, BO can encode a trade-off between probing unexplored regions of the design space ("exploration") and refining previously identified regions favored for the target objective(s) ("exploitation"). This balance between exploration and exploitation ensures scouting of the unexplored design space, inherently minimizing the impact of local optima. This feature also dictates the planning of experiments to meet a certain

objective, avoiding extensive characterization of unfavorable regions. As a result, the overall experimental burden can be reduced, accelerating the optimization. For these reasons, BO has been applied to various applications beyond computer science and robotics such as protein engineering[20–22], reaction optimization[23], synthetic gene design[24–26], material science[27,28], drug formulation[29], and process optimization[30–32]. Some studies have also demonstrated these approaches for designing and optimizing cell culture media[33–35] considering multiple objectives[33] and information sources[34]. These, however, use only continuous design factors in their optimizations.

Here, we demonstrate the application of a BO-based framework to efficiently optimize the composition of cell culture media considering complex design spaces that include both constraints and categorical variables. We illustrate this approach through two distinct use cases relevant to life sciences and biomanufacturing. In one case study, we show the optimization of a media composition that maximizes the viability and maintains the phenotypic distribution of peripheral blood mononuclear cells (PBMCs) ex vivo for up to 72 h. In a second example, we applied this approach to determine a medium to maximize recombinant protein production by the yeast *Komagataella phaffii* (*K. phaffii*), formerly known as *Pichia pastoris* (*P. pastoris*). For both applications, improved performance was achieved compared to current standard media conditions with up to 3-fold reduced experimental burden compared to the state-of-the-art DoE approaches. The reduction of experimental burden was further magnified with the increasing number of factors resulting in a 10- to 30-fold reduction when considering 9 design factors with categorical variables (multiple categories and/or a larger number of levels). We further demonstrate the ability of such a framework to facilitate the transfer of learning and the ability to allow for modifications to the design space a-posteriori such as adding new media supplements. These results show how a BO-based active learning approach to media optimization could improve performance in cell culture for specific objectives efficiently and support additional mechanistic studies on key factors and interactions within these systems.

## Results

The workflow for the BO-based active learning involves both experimental feedback and model training that reinforces the prediction of the target objective (Fig. 1A). The algorithm starts by planning and performing an initial set of experiments to build the first implementation of the surrogate GP model. The GP subsequently interacts with the Bayesian optimizer, which informs the next set of experiments that are designed to balance both exploration and exploitation of the design space. With each new dataset, the GP model is updated, and the process continues until the model converges (or the experimental budget is spent). The studies here focus on optimizing a biological objective (e.g., cell viability, titers) as a function of the composition of media.

### Optimization of media for homeostatic culture of PBMCs ex vivo

PBMCs are a valuable resource and yield data used for drug development, disease monitoring, and therapeutics. Examples include studying drug cytotoxicity[36], co-culturing with solid tumors to understand the role of the tumor microenvironment[37,38], and applications focusing on differentiated subpopulations of the immune cells such as T cells[39] or natural killer (NK) cells[40] for immunotherapies[41]. It is, however, difficult to maintain these primary cells in vitro for extended durations with standard commercial media, as it often leads to reduced viability and shifts in the distribution of cell types. We sought to apply our BO-based approach to perform two sequential optimizations. First, we aimed to determine a media blend of four commercially available medium namely, DMEM, AR5, XVIVO, and RPMI, that would maximize PBMC cell viability. Second, we undertook an optimization using cytokines and chemokines to achieve a balance of key lymphocytic

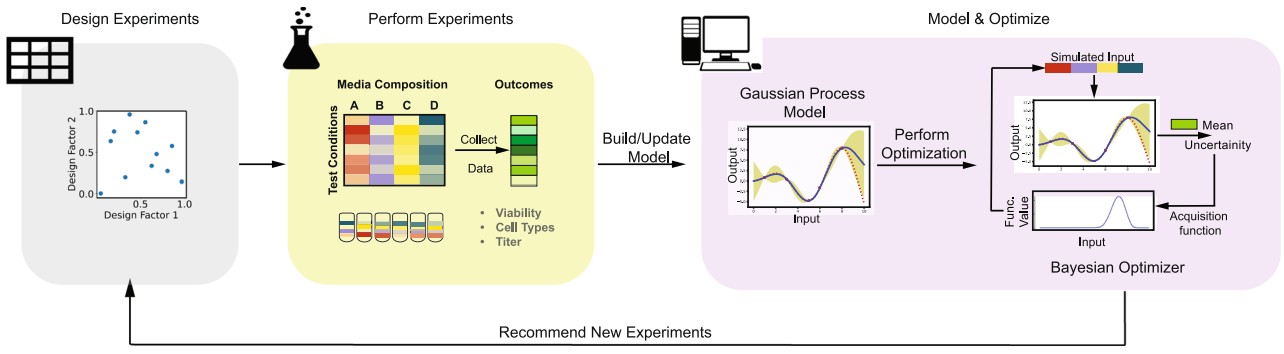

**Fig. 1 | Workflow.** Schematic representation of the Bayesian Optimization (BO)-based iterative experimental design workflow.

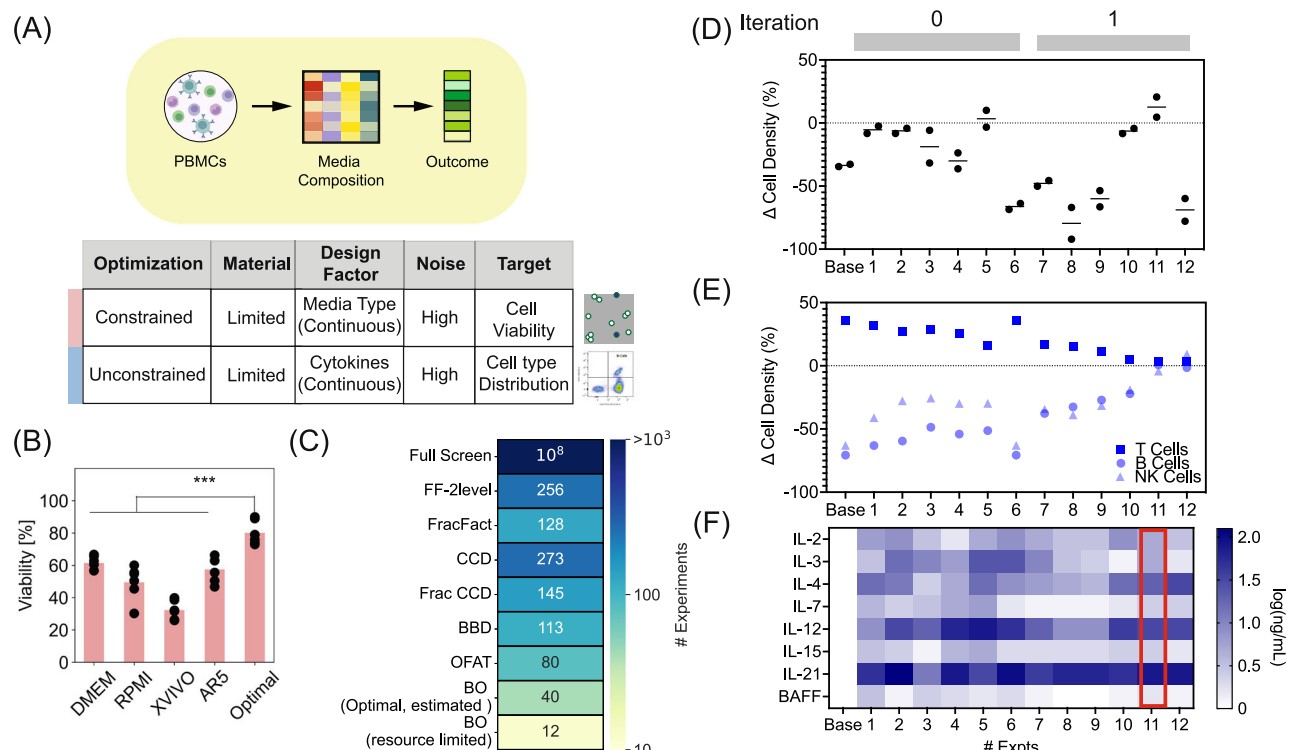

**Fig. 2 | Media blend and cytokine composition optimization PBMC cultures.**
**A** Workflow and study parameters to optimize composition of media to maximize viability and maintain homeostasis of PBMCs in cultures. **B** PBMC cell viability as a function of the optimized media blend compared to individual standard media (DMEM, RPMI, XVIVO, and AR5) represented using data from six biological replicate experiments in each case. Individual double-sided *t*-test was performed between the standard media and optimal media formulation. *** signify a *p*-value < 0.01 with the exact values reported in the Source Data.xlsx file. **C** Comparison of the number of experiments to execute the different strategies for designing experiments considering 8 different cytokines. **D** Change in total cell density before and after 3 days of PBMC culturing using different compositions of cytokines with the red arrow indicating the condition meeting the desired objective. **E** Change in cell density of subpopulations of lymphocytes before and after 3 days in culture under different cytokine compositions. **F** Composition of the cytokines tested in the 12 different experiments (Expts). Red box corresponds to the condition meeting the desired objective (Expt. 11). Source data are provided in Source Data.xlsx file. PBMCs peripheral blood mononuclear cells, BO Bayesian Optimization, OFAT One Factor At Time, BBD Box Behnken Design, CCD Central Composite Design, FF Full factorial, FracFact Fractional Factorial, Frac CCD Fractional CCD, IL Interlukin, BAFF B-cell activating factor.

populations representative of the ex vivo distributions (Fig. 2A). (Alternatively, both the basal media and mixture of cytokines used could be jointly optimized, potentially leading to an improved formulation. This approach, however, introduces a trade-off in iterative optimizations.) By splitting the task into two sequential optimizations, the determined basal nutrient media can serve as a basis for related specific applications involving lymphocytic cell populations (e.g., culturing hematopoietic cancer cells, CAR T cells etc). In this way, only additional optimization of the cytokine/chemokine composition is necessary to modulate the subsequent required properties. This

approach could also allow emulation of the nutrient and signaling environment in vivo in studies to assess the underlying biological mechanisms involved.

The different commercial formulations of media comprise different sets and (or) quantities of nutrients, hormones, and growth factors. We hypothesized that combining these in different ratios could yield a new composition capable of maintaining high cell viability (>70%). This framing yields a constrained optimization problem of continuous design factors with a linear equality constraint, such that the relative contributions of the different media in the blend sums to

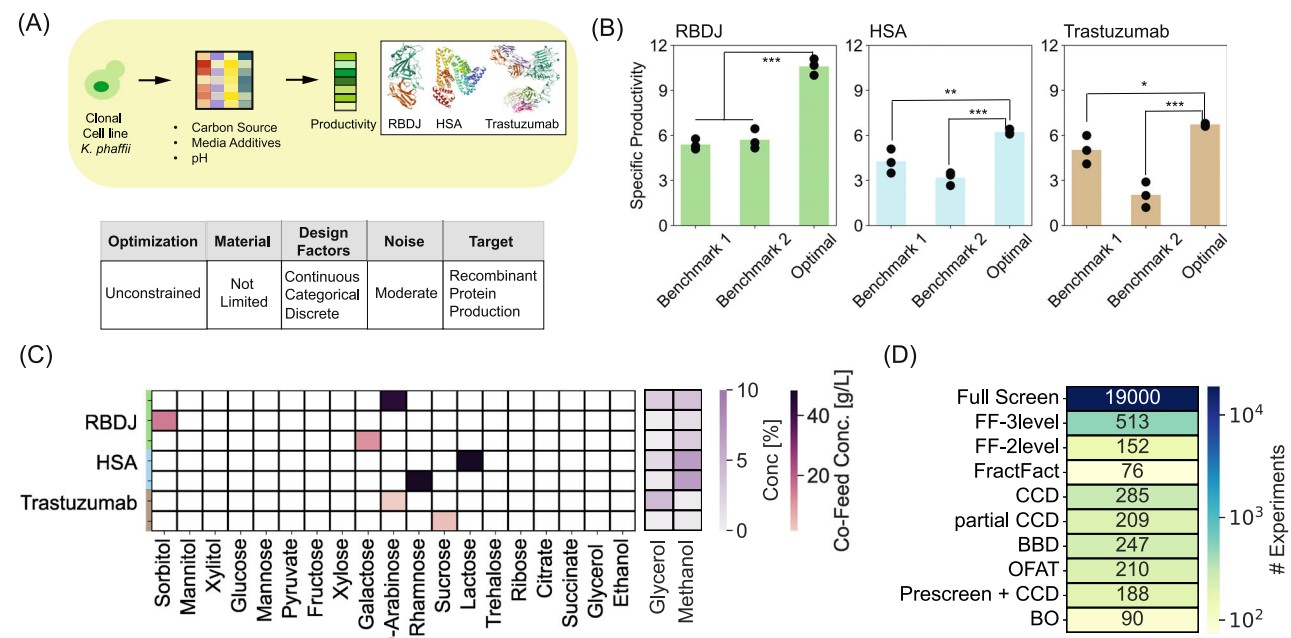

**Fig. 3 | Carbon source optimization for recombinant protein production in *K.phaffii*.** **A** Workflow and study parameters for optimizing media supplements to maximize recombinant protein production in cultures of *K.phaffii*. **B** Comparison of specific productivity using optimized media compositions and 2 benchmark conditions for RBDJ, HSA, and trastuzumab. Individual double-sided *t*-test was performed between the biological replicate experiments performed with benchmark media and optimal media formulation. *, **, *** signify a *p*-value < 0.1, 0.05, 0.01, respectively with the exact values reported in the Source Data.xlsx file. **C** Optimized compositions of carbon sources and specific productivities for the respective proteins. **D** Comparison of the number of experiments to execute the different strategies for designing experiments. Source data are provided in Source Data.xlsx file. BO Bayesian Optimization, OFAT One Factor At Time, BBD Box Behnken Design, CCD Central Composite Design, FF Full factorial, FracFact Fractional Factorial.

100%. We applied our BO-based iterative design approach to maximize the cell viability of healthy PBMCs after 72 h in culture. With 24 total experiments (split into batches of 6 experiments over four iterations, 0 to 3), an optimized blend of the media showed a statistically improved viability of 75–80% compared to individual media which resulted in viability of 60% (Fig. 2B).

Unsurprisingly, we found that the blend of commercial media yielding improved viability of PBMCs did not uniformly maintain the diverse subpopulations of lymphocytes, favoring T cells over populations of B cells and NK cells. To address this imbalance, we selected eight design factors to test based on their known roles in homeostasis, including interleukins (IL-2, IL-3, IL-4, IL-7, IL-12, IL-15, IL-21) and B-cell activating factor (BAFF)[42,43]. The scope of this design space is substantial and would require large numbers of experiments to evaluate using DoE or OFAT (Fig. 2C, see supplementary information for calculation). This trait, therefore, makes performing such optimizations on sparse clinical samples challenging for many applications like drug evaluations in precision medicine or the production of non-T-cell-based cell therapies. Using our approach, we sought to identify combinations of the additional media supplements that would maintain the distributions of cell types after 72 h of cultivation compared to the ex vivo distribution. With as few as 12 experiments (batches of 6 formulations tested in 2 iteration, 0 and 1), we found a combination that retained both the cell density (numbers) and the distributions of cell types (Fig. 2D–F). Interestingly, similar mixtures of the supplements (Expts. 11 and 12, Fig. 2D–F), differing predominantly in the concentration of IL-3, showed a large difference in total cell densities despite similar distributions of cell types as observed ex vivo. The best cytokine combination (Expt. 11, Fig. 2F) also showed reduced concentrations of two cytokines commonly used in media for CAR-T cell cultures (IL-7 and IL-15)[44]. These differences observed in the total composition highlight the nuances of formulating media for primary PBMC cells.

## Enhancing recombinant protein production by *K. phaffii* with carbon source supplements

Optimization of media is also important for applications in industrial biotechnology. As a second example, we applied our BO-based active learning approach to improve recombinant protein production by a yeast host[45], *K.phaffii*, commonly used to produce food proteins, materials, and biologics. Standard compositions of media used for this cultivation rely on either a complex[46] or basal salt media[47] supplemented with glycerol[48] or glucose[49] for biomass accumulation, followed by methanol with or without sorbitol co-feeding to induce recombinant protein production. These formulations (and their use in fermentation) follow canonical standard protocols[50]. We hypothesized that optimizing the concentrations of carbon sources in small-scale cultures could improve the production of secreted proteins. We devised a design space including four design factors: the concentration of glycerol during biomass accumulation, the concentration of methanol during production, the type of carbon co-feed supplement in production (categorical factor), and the concentration of the co-feed. To evaluate the generalizability of our approach and compare convergent solutions, we optimized the required carbon sources for three different proteins with varied complexities: an engineered variant of the SARS-CoV-2 RBD subunit (RBDJ)[51], Human Serum Albumin (HSA), and an IgG1 monoclonal antibody trastuzumab (Fig. 3A). We selected two commonly used carbon conditions as references to compare the resulting compositions (Benchmark1 and Benchmark2 in Table 1).

To account for the categorical variables in our surrogate GP model, we adopted a modified kernel design instead of using OHE. OHE is generally regarded as an inefficient formulation as it increases the dimension of the data and adds sparsity, making the model training inefficient and increasing the data required. To validate the choice, we retrospectively compared the predictive accuracy of the GP models trained with our designed kernel and OHE (Fig. S1A). We confirmed

**Table 1 | Factors considered in the experimental design and the corresponding ranges**

| Factors | Phase | Factor Type | Lower bound | Upper bound | Benchmark1 | Bechmark2 |
|---|---|---|---|---|---|---|
| Glycerol [%] | Outgrowth | Continuous | 0 | 10 | 4 | 4 |
| Methanol [%] | Production | Continuous | 0 | 10 | 1.5 | 1.5 |
| Co-Feed type | Production | Categorical | Ethanol, Glycerol, Sorbitol, Xylitol, Mannitol, Glucose, Fructose, Galactose, Mannose, Trehalose, Lactose, Ribose, Sucrose, Rhamnose, D-Arabinose, Xylose, Citrate, Pyruvate, Succinate | | N/A | Sorbitol |
| Co-Feed Conc. [g/L] | Production | Continuous | 0 | 50 | N/A | 40 |

that the average errors made by GP with the designed kernel were 33–50% smaller compared to OHE. We assessed convergence here as the agreement between the model prediction and the experimental observation (Supplementary information).

We then demonstrated that the BO could identify optimized combinations of carbon sources to improve the specific productivity (measured by mg/L/OD600) relative to the benchmarks, albeit there were different degrees of improvement for each protein (Fig. 3B). These identified media compositions correspond to the best achievable targets for the considered optimization task. In this case, for each molecule, we identified multiple and distinct compositions of media (3 compositions for RBDJ, 2 for HSA, and 2 for trastuzumab) that could optimize protein production to a similar degree (Fig. 3C). We note, however, that given the complex non-linear response surface, using more experiments and time, further iterative learning cycles could yield additional combinations (of the continuous design factors) of media giving similar target values.

These media yielded a 2.5-fold improvement in specific productivity compared to both the Benchmark media for RBDJ (12 mg/L/OD600 vs 5 mg/L/OD600). For HSA, we observed a 2-fold improvement compared to Benchmark2 (6 mg/L/OD600 vs 3 mg/L/OD600) but only a 1.5-fold improvement compared to Benchmark1 (6 mg/L/OD600 vs 4 mg/L/OD600). Finally, for trastuzumab, we observed a 3-fold improvement compared to Benchmark2 (6 mg/L/OD600 vs 2 mg/L/OD600) and no statistically significant improvement compared to Benchmark1 ($p = 0.09$). These differences in observed improvements suggest that there are unique protein-dependent bottlenecks faced by the host that cannot be alleviated by only optimizing carbon sources.

Furthermore, the optimal media compositions differed considerably from the current Benchmark media and among the tested molecules with no two molecules converging to the same composition (Fig. 3C), highlighting the unique requirements faced when optimizing media for the efficient production of different recombinant proteins. The alternative carbon sources considered in this work have not been widely studied for their impact on recombinant protein production in *K.phaffii*. Sorbitol has been used as a co-fed carbon source for generating biomass with *K.phaffii*[52]. L-rhamnose, another carbon source metabolized by *K.phaffii*[53], has shown improved production for HSA here, suggesting a potential alternative to sorbitol. Other carbon sources, such as glycerol, glucose, fructose, and mannose, are known to support growth/biomass accumulation while having a repressive impact on the pAOX1 promoter[52,53], making them unsuitable candidates to promote protein production. Many carbon sources not known to be metabolized by *K.phaffii*, including D-arabinose, D-ribose, D-xylose, galactose, lactose, xylitol and sucrose[53] interestingly showed benefits as a co-feed to enhance recombinant protein production here. Which carbon sources were beneficial, however, depended on the protein produced. How these different carbon sources influence protein production would merit further investigations to assess their influence on the cellular states.

Given this trait, it is ideal to develop new protein-specific media without requiring excessive resources or time. Using our BO-based active learning approach, we found we could optimize the carbon sources required using only 90 experiments over 7 experimental iterations (23 experiments in the initial design followed by 11 experiments each in the future iterations), requiring a total of ~1–1.5 months. This total experimental load was ~2.5–3 times lower than the predicted number of experiments required for standard designs of DoE (Fig. 3D, see supplementary information for calculation) and several orders of magnitude lower than an exhaustive search to fully screen the design space, assuming a grid of 10 levels (Fig. 3D). We acknowledge the full-screen design is infeasible in most practical settings. Since the BO algorithm considers the design space as a continuum (instead of discretized space used in most statistical DoEs), the full screen provides

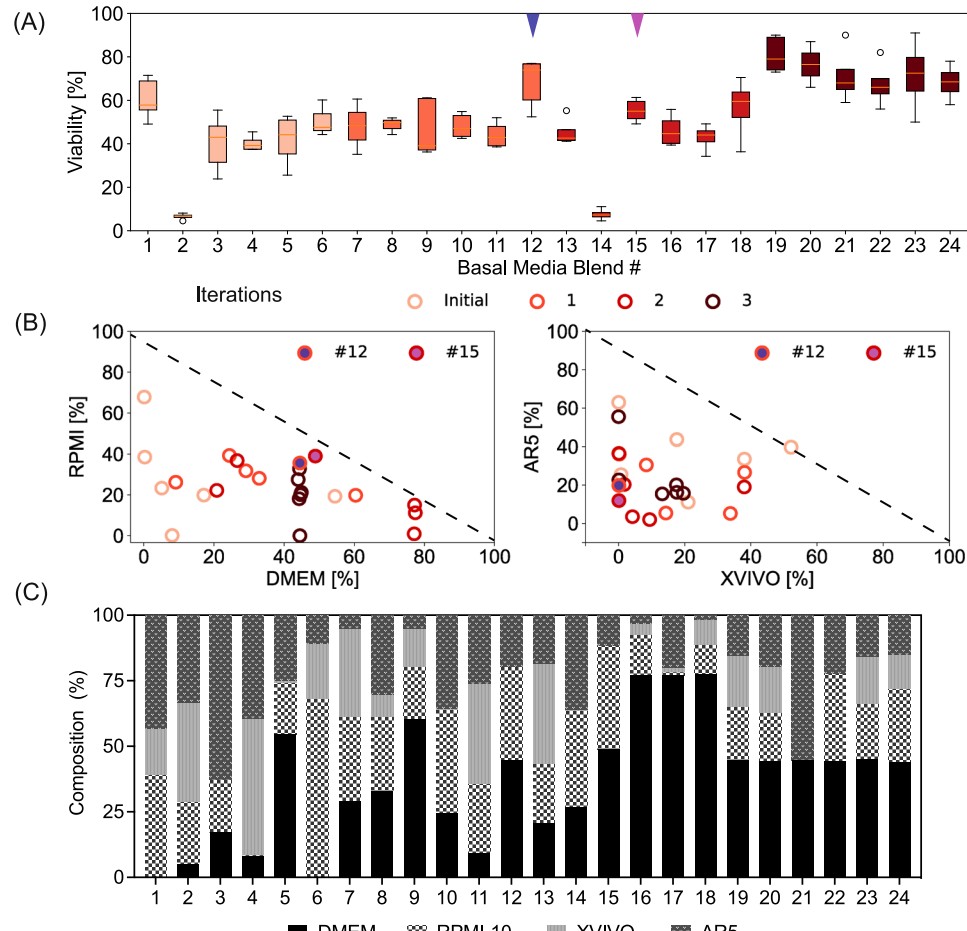

**Fig. 4 | Characterization of exploration-exploitation trade-off for PBMC media blending case. A** Evolution of cell viability of the experiments planned in the different iterations indicated as boxplots of the biological triplicate data (with some experiments having up to five replicates). The box spans between first and third quartile encompassing a line for the median. The whiskers extend to the farthest data point that falls within 1.5 times the interquartile range. Points outside this range is indicated as fliers. The progressively increasing shades of red correspond to the increasing iteration number. **B** Evolution of the location of the experiments in the feasible design space during the different iterations represented in a pairwise plot of the design factors. **C** Composition of the different media blends tested in experiments planned in the different iterations. Source data are provided in Source Data.xlsx file.

an estimate of the spread for the complete design space (approximated with a grid of 10 levels here) (see Supplementary Information).

**Elucidation of the algorithm characteristics: Exploration-Exploitation tradeoff**

Having demonstrated the capabilities of the BO algorithm for accelerated, resource-efficient media optimization, we next sought to elucidate the characteristics of the algorithm by investigating how the algorithm navigates the selected design space and its corresponding impact on the target. First, the use of a space-filling design to generate the initial iteration of experiments maximized the variability in the input-output combinations seen by the GP model, thus, allowing for an efficient initial representation of the system by the model. The use of such space-filling designs to generate initial data has shown success in several applications[29,30], including media optimization[33,34]. The wide coverage of the design space tested in the initial iteration for the PBMCs (Fig. 4B) resulted in a wide range of cell viabilities measured (from 5 % to 62%) (Fig. 4A). The diversity of the initial assessment is also explicitly evident from the varied ratios of the different media types (DMEM, AR5, XVIVO, and RPMI) in the designed initial blends (Fig. 4C). We note that the feasible region to plan experiments in this example is a non-cuboidal design space defined by the constraint imposed on media blending. That is, the sum of the ratios of different media type should equal one (Black

dashed line, Fig. 4B). Similarly, the initial space-filling design to optimize the carbon sources for the yeast cultivations was also broadly distributed (Fig. 5B) subsequently resulting in a wide distribution of specific productivities measured (ranging from 0.1 to 9 mg/L/OD600) (Fig. 5A).

In each future iteration, the optimizer attempted to find a trade-off between planning experiments in unexplored regions of the design space and refining its confidence in regions identified as favorable for the target objective (exploitation). Practically, given that the unexplored regions represent a larger portion of the design space at the start of the sequential campaign, we confirmed that the recommended experiments planned in early iterations of the optimization favored exploration, and then progressively moved towards exploitation-dominated designs, narrowing down the probed parts of the design space.

For the example of media blending for PBMCs, we observed in the pairwise plot of design space (Fig. 4B) as well as the univariate plot of individual design factors over iterations (Fig. S4), that exploration dominated the first two iterations (Iterations 0 and 1), resulting in cell viability varying from 5% to 75% over (Fig. 4A). Subsequently, Iteration 2 included a mix of exploration and exploitation: For instance, Blend 15 exploited a previously observed region covered by Blend 12 (Fig. 4B, C; Fig. S4). Subsequently, the final iteration (Iteration 3) exploitatively reduced the search space to a specific ratio of DMEM with a focus on

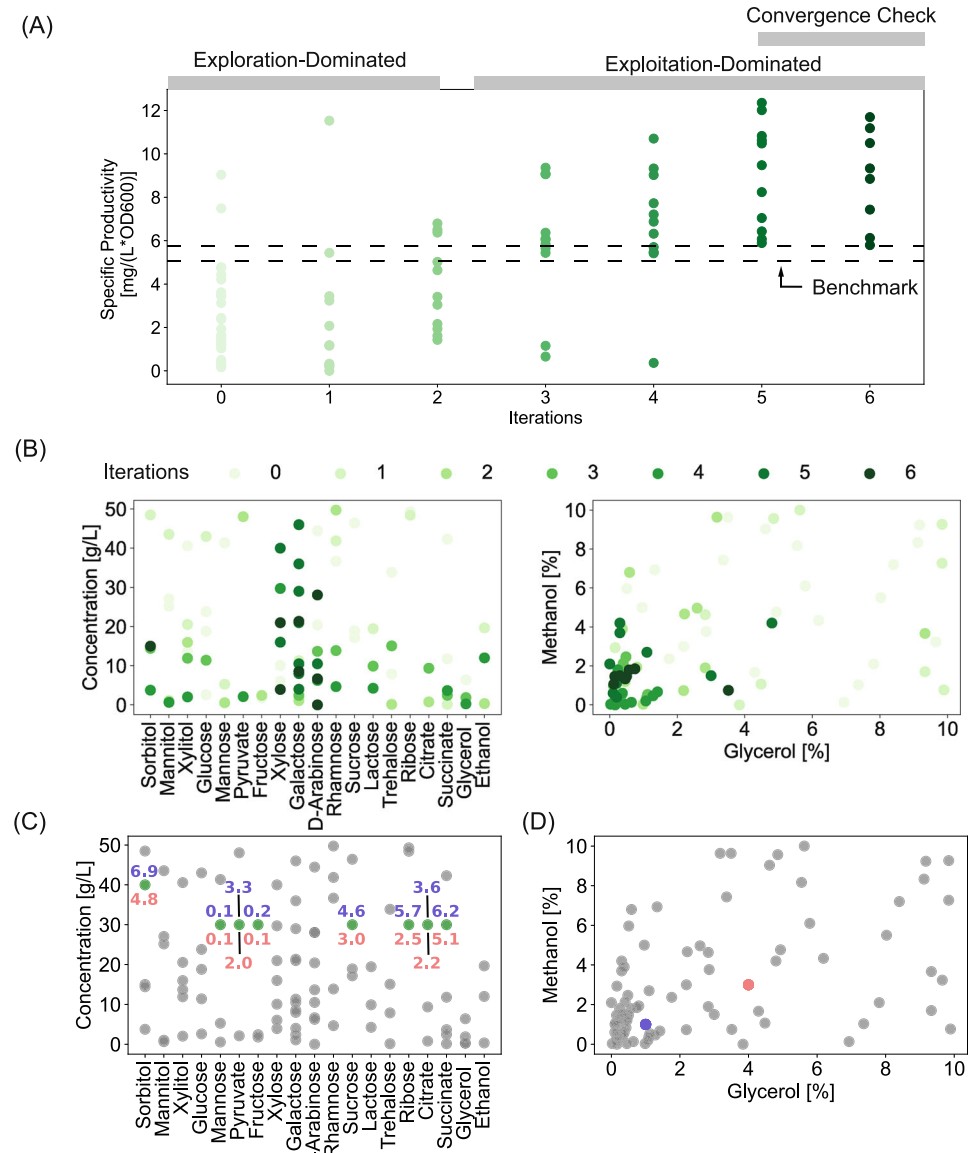

**Fig. 5 | Exploration-exploitation trade-off and confirmation of poor specific productivity in the region lacking experiments. A** Evolution of specific productivity distribution of the experiments planned in the different iterations. The progressively increasing shades of green correspond to increasing iteration numbers. **B** Evolution of the location of the experiments in the design space in the different iterations represented in a pairwise plot between the design factors. **C** Scatterplot of the carbon source type and concentration of the experiments tested (Grey circles) along with the validation experiments planned (Green Circles). The numbers represent the specific productivity with the colors indicating the corresponding glycerol–methanol conditions in D (**D**) Scatterplot of the percentages of methanol and glycerol tested (Grey circles). Pink and purple circles–two combinations of glycerol and methanol tested with the selected carbon source type and concentrations marked in Fig. 5C. Source data are provided in Source Data.xlsx file.

perturbations involving different combinations of the other media types (Fig. 4B, C; Fig. S4).

For the optimization of media for protein production with yeast, using the example of RBDJ, we similarly observed a large distribution of values of specific productivity (Fig. 5A) with widespread spacing of experiments covering the design space in Iteration 1 and 2 (Fig. 5B). Unsurprisingly, the algorithm planned experiments in Iteration 1 for types of carbon sources not probed in the initial experiments, as observed in the pairwise plot of the design space (Fig. 5B) and the univariate plot of the individual design factors (Fig. S5) over the different iterations. This outcome can be attributed to the objective function being dominated by the higher uncertainty manifested in these regions of the design space. Iterations 3 through 6, however, emphasized exploitation as observed by the higher fraction of experiments planned in a limited region of the design space, resulting

in increased specific productivity (Fig. 5A) with limited testing of other regions of the design space (Fig. 5B). Particularly, the favorable part of the design space corresponding to the continuous variables, glycerol and methanol, were identified in Iterations 3 and 4 (Fig. S5A, B). For the categorical variable (co-feed type) and the categorical-coupled continuous variable (co-feed concentration), however, Iterations 3 and 4 focused on navigating different carbon sources at a limited range of concentrations, while iterations 5 and 6 limited the co-feed type to the most favorable ones, exploring a range of concentrations for these experiments (Fig. S5C, D).

These analyses show the algorithm progressively learned the favorable/unfavorable regions of the design space as it reduced the experimental testing in the unfavorable regions. For the PBMC culturing, this progression resulted in a few experiments being planned for media blends with XVIVO > 60%. Reviewing the tested media

**Table 2 | Additional factors considered in the experimental design of the transfer learning study and the corresponding ranges**

| Factors | Phase | Factor Type | Lower bound | Upper bound | Benchmark |
|---|---|---|---|---|---|
| Glutathione [mM] | Outgrowth, Production | Continuous | 0 | 50 | 0 |
| pH [-] | Outgrowth/ Production | Discrete | 5.75, 6.0, 6.5 | | 6.5 |
| Tween20 [%] | Outgrowth, Production | Continuous | 0 | 10 | 0 |

blends, it is clear that blends dominated by higher XVIVO amounts (Blends 2, 4, 11, and 13–Fig. 4C) resulted in lower cell viabilities with the best viability of only 40% (Fig. 4A).

Similarly, for the production of RBDJ by *K.phaffii*, specific carbon source types (e.g., mannose, pyruvate, ribose, glycerol, etc.; especially in the concentration range of 20−40 g/L) were restrictively probed (Fig. 5C) and were not selected in the optimal media formulation. To validate that these ignored parts of the design space would result in poor specific productivity, we manually picked 16 different conditions with the eight different carbon sources from these regions of the design space and tested the specific productivity of RBDJ under those conditions (Fig. 5C). The eight carbon sources were tested with two different glycerol and methanol concentrations (Fig. 5D). The specific productivities for all these selected conditions ranged from 0.1 to 6 mg/L/OD600 and were lower compared to the algorithm-identified optimal conditions for RBDJ (specific productivity of 12 mg/L/OD600) (Fig. 3B).

Taken together, these analyses on the iterative progression of the models for both cases tested support the ability of BO to accelerate optimization and minimize resources used. The synergy between experimentation, model building, and optimization in this iterative framework, coupled with the gradual trade-off between exploration and exploitation in each iteration reduces the number of experiments allocated to less optimal regions of the design space.

### Extending the approach with Transfer Learning to incorporate additional media supplements

Both cases tested here yielded improved performance for the respective objectives subject to the selected media components. It is apparent in both cases and, more generally, that performance could be further enhanced by considering additional influential factors in the optimization. It is, therefore, desirable to allow continued improvement and extensions of the model to incorporate new design factors and (or) objectives. In these cases, the ability to strategically use the learnings generated from the existing data would be crucial to minimize the resources and experiments required for future optimizations. This goal requires capabilities to transfer learnings to modified design spaces (e.g., expanding the range of the design factors or adding additional factors) or to alternative biological systems (e.g., production of other molecule types or culturing blood cancer cell lines). In this work, we explored the feasibility of extending our current framework to the first case, that is, to transfer learning for modified design spaces. GP models are well-suited to this extension since they can learn from data through the context of similar experiments in the design space. We sought to demonstrate this feature by considering five additional media additives to optimize the recombinant protein production in *K.phaffii* (Table 2), using HSA as a test protein because it showed only moderate improvement when only the selection of carbon sources was considered.

To seed this new iteration of the model that included the additional supplement, we used the current GP as the prior (instead of using a space-filling design) and used the optimizer to determine subsequent experiments. We included all nine factors (the new supplements and the four prior ones) in designing new experiments, allowing the model to re-learn dependencies in the modified design space as needed. Iterating in this way yielded a modified composition of media that improved the specific productivity for secreted HSA

from 6 mg/L/OD600 (starting point) to 13 mg/L/OD600 (Fig. 6A). This realized improvement corroborates our hypothesis that hosts producing recombinant heterologous proteins may require specific tailored media composition to maximize their specific productivity due to unique metabolic requirements or protein-dependent features (folding, assembly).

One impact of starting the model with the GP from the prior task is that the algorithm minimized the experiments planned for certain carbon sources that had yielded poor specific productivities previously (e.g., sorbitol, mannitol, xylitol, glucose, mannose, succinate, and glycerol), and focused on a subset of alternative carbon sources within the first two rounds (Fig. S6). Similarly, starting from the initial iteration, the algorithm planned most of the experiments with concentrations of glycerol <5% and those of methanol between 1.5 and 8%. As a result, new optimized media conditions required only 72 additional experiments (including 12 to confirm model convergence) to consider the new design space of 9 factors. This ability to use prior learning in the form of a surrogate model thus resulted in at least a 20% reduction in the experiments that started from scratch using the BO approach for the new design space which would have required at least 90 experiments (based on the carbon source optimization case). The total number of experiments from the two tasks together was about ~160 experiments–several orders of magnitude lower than a practically infeasible full screen and ~10−30 times fewer experiments than traditional DoE approaches (Fig. 6B).

The experimental burden of DoEs increases substantially as the design space expands, and consequently, often only a subset of factors are considered in any given optimization to reduce the overall number of experiments performed[10]. For instance, in our nine-factor study here, a study might hold the prior optimized conditions for the carbon source constant and simply perform a separate DoE for the five additional design factors. Interestingly, in the case we considered, however, this common approach would not have yielded the best productivity: The new media supplements resulted in an alternative preference of carbon sources for maximizing the productivity (from rhamnose and lactose to ribose, D-arabinose, and galactose) (Figs. 3C, 6C). Whereas the use of rhamnose as a co-feed resulted in high specific productivity (~10.5 mg/L/OD600), the concentration was altered from that found with the four-factor design space. These data support the flexibility of this BO-based active learning approach for media optimization to accommodate the posterior addition of design factors (such as new additives) or expand the concentration range of existing design factors[29]. This 'bootstrapping-type' approach to optimization would become increasingly valuable as the design space expands, as is often the case in media optimization when considering several types and concentrations of additives.

### Discussion

In this study, we have presented an accelerated and resource-efficient approach for media development using Bayesian optimization-guided iterative experimental design. Using two unique experimental systems, we have shown its capabilities for cell cultures used in common applications in both life sciences and biomanufacturing. First, we optimized the media composition to maximize viability and maintain homeostasis of PBMCs in culture. In the second case, we optimized the concentrations of additional supplements used in cultivation media for *K.phaffii* to maximize the production of three different

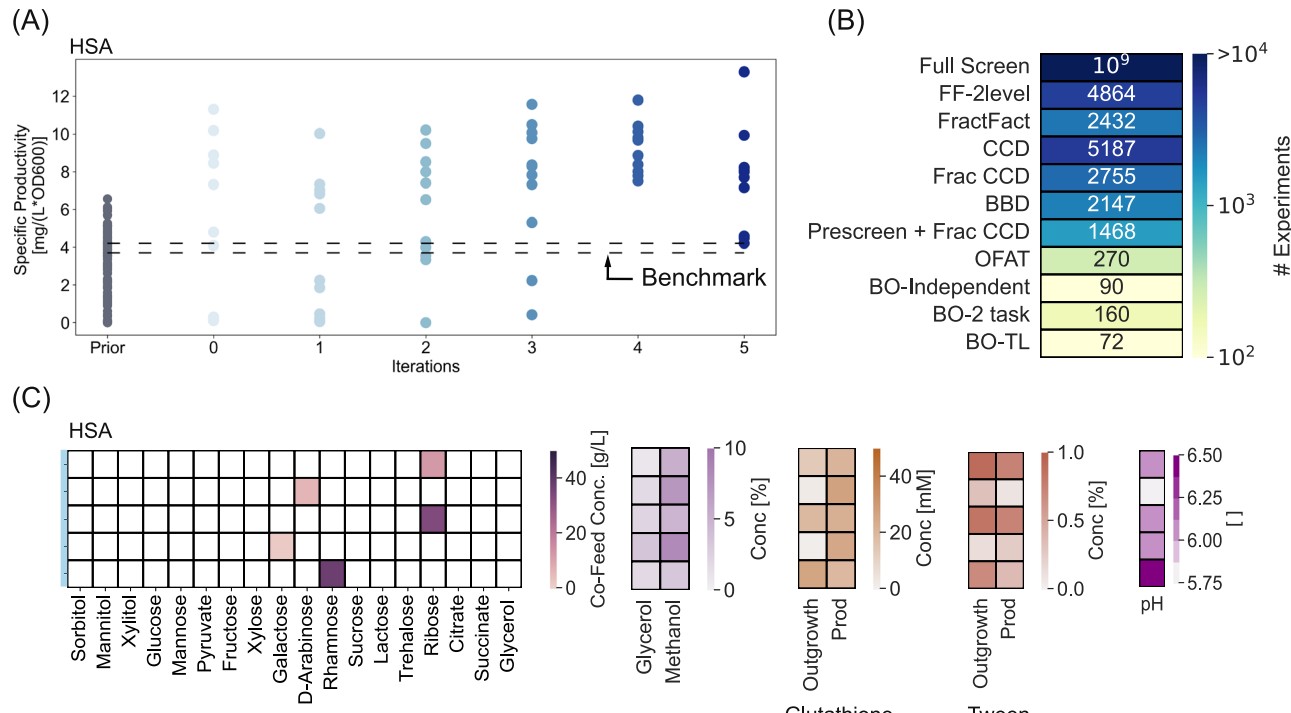

**Fig. 6 | Transfer learning to a new design space with additional factors.**
**A** Evolution of specific productivity distribution of the experiments planned in the different iterations. The progressively increasing shades of blue correspond to the increasing iteration number. **B** Comparison of the number of experiments to execute the different strategies for designing experiments. **C** Composition of media supplements for top 5 candidates with highest specific productivities in decreasing order. Source data are provided in Source Data.xlsx file. BO Bayesian Optimization, OFAT One Factor At Time, BBD Box Behnken Design, CCD Central Composite Design, FF Full factorial, FracFact Fractional Factorial.

recombinant proteins. In both cases, improved performance was achieved compared to current standard media conditions with at least a 3- (e.g., 4 design factors) to upto 30-fold (e.g., 9 design factors with consideration of multiple categorical variables and/or a large number of categories) reduced experimental burden compared to the state-of-the-art DoE approaches. Inferring from the selection of experiments within the design space during iterative rounds, it is evident that this efficiency results from the capability of the algorithm to focus the experimental effort on identifying favorable regions for the targeted objective via a tradeoff between exploration and exploitation, and therefore, minimizing the experimental efforts in undesirable regions, reducing the overall time and resources required.

The examples of optimizations performed here, including the incorporation of transfer learning to extend the design space, show the potential for this BO-based active learning strategy to minimize experimental costs and time for complex biological tasks like media optimization. The extensibility of the models makes it possible to add new design factors or objectives without introducing artificial constraints or biases needed to manage the budget for experimental exploration of the large design space. GPs used for the models intrinsically provide an efficient way to accommodate additional media supplements or objectives a-posteriori since they associate unexplored regions of the design space (the influence of new added factors/goals) with a larger uncertainty. This approach should provide benefits for adapting models to new systems or tasks where the input materials available for experiments impose a natural limitation on how many experiments are feasible during optimizations (e.g., development of primary cancer cell culture from tissue biopsies, pediatric cell therapies).

This approach derives its advantage from coupling data collection (design), modeling, and optimization into a comprehensive, iterative process, thus strategically navigating the design space based on experimental feedback. In contrast, DoE approaches offer static designs irrespective of design factors or target types, with no active accumulation of learnings based on data collected from the experimentation. Additionally, our approach offers more generalized capabilities that can account for different types of design factors and optimization tasks compared to DoEs, which can only work with discrete or continuous and cannot generate designs for constrained spaces or for categorical variables with linked continuous variables. Finally, the approach here offers broader coverage of a design space compared to DoE studies that discretize the design space and only test the corner and center points of the design space in a defined campaign. The case studies presented here together invoked a range of scenarios that account for different types of design factors (continuous, discrete, categorical), frameworks for optimization (constrained, unconstrained), degrees of biological noise (moderate to high), and intrinsic limitations on the material (cost, time).

One disadvantage of our current implementation is the procedure for in silico sampling needed to accommodate categorical variables relies on randomly sampling many potential experimental suggestions, increasing computational time (~2–4 min/experiment). Using recent advancements in optimization approaches[54], the efficiency of this sampling process could be improved. Furthermore, the space-filling designs used can be modified to account for constrained design spaces and categorical variables[55–59], thus, improving the efficiency of initial design and iterative-sampling. Additionally, the current implementation does not account for parallelization or utilize GPU acceleration, both of which could significantly enhance computational efficiency. Further advances to enhance the computational efficiency using sparse Gaussian Process regression could also reduce the memory required, and enable efficient processing of larger numbers of factors and sizes of datasets[60].

In conclusion, we have focused here on a BO-based active learning approach to media optimization and demonstrated improved performance in cell culture for specific objectives. The scalability of the methods here, however, also depends on practical considerations in media optimization, such as the cost, solubility, stability/cross-compatibility, and environmental impact of the components. These considerations could be included in the optimization as additional objectives and (or) constraints and addressed through further advancements of the current framework.

The examples presented here also highlight the potential for this approach to extend to other complex biological applications, such as process development. The extensibility of the strategy suggests that incorporating such BO-based experimental design as a standard practice in life sciences research could facilitate both the generation of foundational data to support new predictive models in biological systems and further accelerate development pipelines for new systems. For example, the models developed here to optimize media to sustain ex vivo PBMC cultures could be extended to support the establishment of patient-derived cell lines in cases where the material is limited, such as for needle biopsies or rare/pediatric cancers. Similarly, media and process development for new protein molecules or engineered strains could be accelerated by starting from the models and data generated here. Related applications like scaling up process conditions across systems could also benefit from this strategy for optimization which uses synergy between iterative experimental data and modeling to inform optimization. We postulate that this approach and similar ones like reinforcement learning could establish modular frameworks for improving predictive capabilities across multiple complex biological systems.

## Methods

### Culturing PBMCs

Peripheral blood mononuclear cells (PBMCs) were thawed in a 37 °C water bath, followed by adding the thawed cells to 4 mL of DMEM (Gibco™, Cat#11965092) supplemented with 10% fetal bovine serum (FBS; Sigma-Aldrich, Cat#F4135). The cells were centrifuged at 500 x g for 5 min, the supernatant was discarded, and the pellet was resuspended in 2 mL of fresh DMEM. Cells were counted, and the appropriate volume of cell suspension was calculated to achieve a final concentration of $1 \times 10^6$ cells/mL in 2.3 mL of the chosen reactor medium (determined based on BO suggested experimental design). After centrifugation at 500 x g for an additional 5 min to remove any residual DMEM, the cells were resuspended in 2.3 mL of the reactor medium. A volume of 200 μL of the cell suspension was then pipetted into each well of the specialized C.BIRD™ reactor plates (CYTENA, Cat#3000012), ensuring that empty wells were filled with 200 μL of PBS or water and 10 mL of PBS (Gibco™, Cat#10010023) or water was added to the plate reservoir. The C.BIRD™ reactor plate lid was attached to the 96-well plate, and the assembly was placed in a 37 °C incubator with standard settings used for cancer cell culture, including 5% $CO_2$ and high humidity (~95%) to maintain pH and prevent evaporation. Basal media included either DMEM (Gibco™, Cat#11965092) or RPMI 1640 (Gibco™, Cat#11875093), each supplemented with 10% FBS (Sigma-Aldrich, Cat#F4135) and 1% Penicillin-Streptomycin (Gibco, Cat#15140122). AR5 and XVIVO media were prepared according to manufacturer instructions (AR5: CellGenix, Cat#20807; XVIVO 15: Lonza, Cat#04-418Q). Detailed preparation protocols are available at cellfactory.broadinstitute.org/#/sops. 6-8 biological replicates were performed for different designed experimental conditions. For viability measurements (PBMC – Media blending case study), the contents of two replicate cultures were pooled, thus resulting in three – four replicate readings of the viable cell count. An average of the replicate readings was used as the modeling target. During the cytokine optimization, the content of the replicate cultures was pooled into a single pool that was used for flow cytometry measurements to quantify both the cell viability (using viability dye) and the lymphocytic subpopulation of interest.

### Viability Assays

Cell viability and counting were performed before and after incubation in the C.BIRD™ microplate bioreactors (CYTENA). After the culture duration (72 h), cells from replicate conditions were pooled into a single tube, and 40 μL of EDTA (Thermo Fisher Scientific, Cat#AM9260G, 0.5 M stock) was added to each sample, to help prevent aggregation and detach PBMCs from the reactor surfaces to ensure accurate cell counting. A 1:1 mixture of the cell suspension and AOPI (ViaStain AOPI; Nexcelom, Cat#CS2-0106) was prepared, and cell counts were performed in replicates (three or four) using the Nexcelom Cellaca plate reader.

### Flow cytometry

Peripheral blood mononuclear cells (PBMCs) were isolated from healthy donor blood samples (STEMCELL Technologies, Cat#70025) using density gradient centrifugation and pooled into a 1.5 mL Eppendorf tube. A 100 μL aliquot of cells was extracted for viability assessment, while the remaining volume was centrifuged at 500 x g for 8 min. During centrifugation, 5 μL of Human TruStain FcX™ Fc receptor blocking solution (BioLegend, Cat#422302) was added to 95 μL of FACS buffer (phosphate-buffered saline (PBS) supplemented with 1% bovine serum albumin [Sigma-Aldrich, Cat#A4503] and 0.1% sodium azide [Sigma-Aldrich, Cat#S2002]) per sample.

After centrifugation, the supernatant was discarded, and the cells were resuspended in 100 μL of Fc block solution and incubated on ice for 20 min. Staining was performed using the following antibodies, each diluted 1:20 in FACS buffer unless otherwise noted: (i)Anti-CD20-FITC (BioLegend, Cat#302304) – 5 μL/test, (ii)Anti-CD45-APC (BioLegend, Cat#304012) – 5 μL/test, (iii) Anti-CD56-APC-Alexa Fluor 750 (BioLegend, Cat#318342) – 5 μL/test, (iv) Zombie Violet Fixable Viability Dye (BioLegend, Cat#423113) – 1:500 dilution in PBS, 1 μL/test.

Cells were incubated on ice in the dark for 30 min. Following incubation, the cells were washed twice by centrifugation at 500 x g for 5 min and resuspended in 200 μL of FACS buffer. Flow cytometric data acquisition was performed on a Beckman Coulter CytoFlex LX, utilizing appropriate laser configurations and voltages for each fluorophore. The flow gating strategy is summarized in the supplementary information.

### K.phaffii cultivation

Cultivations with different media formulations were tested on a plate scale. Experiments were performed in Axygen twenty-four well deep well square plates (total volume 10 mL, working volume of 3 mL) at room temperature on Benchmark Orbi-Shaker™ plate shakers (600 rpm). Complex media commonly known as BMxY – 1.34% nitrogen base w/o amino acids (Difco Yeast Nitrogen Base w/o Amino Acids, Cat# 291920), 1% yeast extract (Difco™ Yeast Extract, Cat# 210929), 2% peptone (Bacto™ Peptone, Cat# 211677), potassium phosphate buffer - at the set pH was used as the basal media for the cultivation. Appropriate types and concentrations of additives were added to this basal media according to the experimental plan generated by the algorithm for both the outgrowth and production phases of the cultivation. All additives were obtained from Sigma-Aldrich. Cultivations were inoculated at 0.1 OD600 from working cell banks, grown for 24 h, pelleted, and resuspended in fresh production media to induce recombinant gene expression. Supernatant samples were collected after 24 h of production, filtered, and analyzed using Agilent Infinity 1260 High Performance Liquid Chromatography to quantify titer. Reverse phase column (Agilent Technologies, Cat# PL1612-1801) was used to quantify titers for RBDJ and HSA molecules while Biomonolith protein A column (Agilent Technologies, Cat#5190-6903) were used to quantify trastuzumab titers.SDS-PAGE (Invitrogen Novex™ 12% Tris-

Glycine Plus Midi Gel, Cat# WXP01226BOX) was carried out as described previously[61] to confirm protein bands of the right size and no product-related variants. Specific productivity was defined as relative titer normalized by cell density, measured by OD600 on Tecan Infinite M Nano$^+$ plate reader. Biological duplicates were run for all designed experiments, and the average value of the biological duplicate was considered for the modeling. Cultivation using media indicated in Benchmark 1 was run as a control experiment on each iteration. The process noise was calculated as the variance across the control experiment for all instances including the contemporaneous iteration. For building the initial model, the variance was calculated from a preliminary experimental campaign to select molecules expressed with Benchmark 1 media. This estimated process noise was incorporated into the modeling (see Gaussian Processes).

### Initial design

A Latin hypercube sampling (LHS) was used to generate the initial design for the continuous variables while a uniform random design was used for the categorical variables. In LHS, the designs are generated such that each hyperplane of the design space has only one point in contrast to purely random sampling from a uniform distribution that results in uneven spacing of the experiments in the design space[3,29,30]. However, since LHS considers continuous unconstrained design spaces, random sampling was used for constrained design spaces (PBMC media blending study) and categorical variables (for the *K.phaffi* recombination protein production optimization case study). For constrained design spaces random design within the feasible design space was used as the initial design implemented using the GpyOpt python package. We note here that the choice of the initial design is expected to impact the number of iterations required for the convergence of the algorithm.

### Gaussian processes

Gaussian processes (GPs) are probabilistic models that learn an underlying unknown black-box function (i.e., the relationship between media additive and specific productivity) by representing them as a distribution of functions. This distribution of function is characterized by a mean $m(x)$, and a covariance function $k(x,x')$ that is dictated based on prior beliefs about the system (Eq. 1).

$$f(x) \sim GP(m(x), k(x,x'))$$ (1)

The covariance function is defined through the kernel which encodes the similarity between points in the design space and the selection of this function depends on the beliefs about the smoothness, periodicity, and trends in the design space. Since smooth trends are expected over the continuous space in this application, we used a smooth flexible Matern kernel.

$$k_{cont}(x, x') = \frac{2^{1-\nu}}{\Gamma(\nu)} \left( \frac{\sqrt{2\nu}|x - x'|}{\theta} \right)^{\nu} K_{\nu} \left( \frac{\sqrt{2\nu}|x - x'|}{\theta} \right)$$ (2)

x indicates the continuous inputs, $\Gamma$ is the gamma function and $K_\nu$ is the modified Bessel function. The $|x - x'|$ indicates the distance between the two points. The kernel value decreases as the distance increases beyond the length scale, $\theta$. $\theta$ is a vector with the same dimension as the number of design factors. $\nu$ is the smoothness controlling parameter with a higher value of $\nu$ implying a smoother function. Common values of $\nu$ for this kernel are 1.5 and 2.5. The first value yields once differentiable functions and the second results in twice differentiable functions. The categorical overlap kernel (Eq. 3) is compatible with at least twice differentiable kernels such as Radial Basis Function (RBF) or Matern ($\nu = 2.5$)[62], motivating the selection of the kernel. For the PBMC case study (considering only continuous design factors) with higher noise, a Matern kernel with $\nu = 1.5$ was used

(We note that both $\nu = 1.5$ and 2.5 performed similarly for these data (Supplementary Information, Fig. S2)).

For the definition of the categorical variable, we use the formulation of the categorical overlap kernel suggested by Ru et al.[62], which defines the kernel as the total number of categories that overlap between the two points $h_i$ and $h_i'$.

$$k_{cat}(h, h') = \frac{\sigma}{c} \sum_{i=1}^{c} I(h_i - h_i')$$ (3)

h indicates the categorical inputs, and c represents the total number of categorical variables considered (in this case, 1 - that is the type of carbon source).

The final kernel over the categorical-continuous variables together ($z$) is also adapted from Ru et al.[62], as follows:

$$k(z, z') = \alpha * (k_{cat}(h, h') * k_{cont}(x, x')) + (1 - \alpha) * (k_{cat}(h, h') + k_{cont}(x, x'))$$ (4)

To this kernel, process noise was added through a white noise kernel with a fixed variance computed based on the variation in the replicates of the control experiment for each iteration/across different iteration.

The length scale in the continuous kernel and the trade-off parameter ($\alpha$) in the mixture kernel is a hyperparameter that is updated by maximizing the marginal likelihood based on data, refining the priors and resulting in the posterior distribution.

The length scales were initialized to the default settings in GPy and the trade-off parameter (where relevant) was initialized to 0.5 giving equal weight to both additive and multiplicative terms at the beginning. With each iteration, the training errors were monitored through RMSE in training to ensure the success of parameter optimization. Using *K.phaffii* as an example, we verified the stability of this approach to initialization a-posteriori (Supplementary information, Fig. S3).

The noise addition to function estimation ($\epsilon_i$) is provided through the likelihood function, in this case, Gaussian likelihood (Eq. 6).

$$y_i = f(x_i) + \epsilon_i$$ (5)

$$\epsilon_i = \mathcal{N}(0, \sigma_i^2)$$ (6)

Gpy package was used to set the kernel and the Gaussian process implementation in Python.

### Bayesian optimization

The GP is then used by an optimizer that suggests the next set of experiments (media conditions) using an acquisition function that encodes a tradeoff between characterizing previously unexplored parts of the design space and exploiting the regions with promising targets (higher specific productivity). To determine the PBMC basal media blend determination, cell viability was used as the target property whereas for the cytokine optimization an aggregated objective function was used to consider both cell viability and cell differentiation (Eq. 7). Finally for the case study with *K.phaffii*, specific productivity was used as the target.

$$Target = Viability\,factor * \sum_{j=\{NK, T, B\}} Frac_j$$
$$if\ Viability \le 1 : Viability\,factor = Viability$$ (7)
$$if\ Viability > 1 : Viability\,factor = \frac{1}{Viability}$$

In this work, for optimization problems with only continuous variables, we use an upper confidence bound acquisition function compared to other alternatives that define the tradeoff simply using the predicted mean and uncertainty. For continuous unconstrained

optimization problems (e.g., the cytokine optimization for PBMC culture) a local optimizer such as LBFGS is used to maximize/minimize the acquisition function, and a trust-region-based algorithm is used for constrained optimization (PBMC basal media blend). These optimizers are implemented through the scipy package in Python. In the presence of categorical, continuous optimization (*K. phaffi* cultivation media optimization), for the first few iterations (Iterations 0 to 3), we used the multi-armed bandit-based sampling of the categorical variable adapted from Ru et al.[62], and Thompson sampling for the continuous variable. For the subsequent iterations (Iteration 4 to 6) a brute force approach and simulated 10000 points via LHS, and the points with the maximum/minimum acquisition function were picked.

Finally, the original implementation of BO is a truly sequential design approach, planning one experiment in each iteration[16]. For most applications, as in our case, it is more practical to perform several experiments in parallel. Thus, we use a variant of BO called batch BO using the "constant liar" approach using the mean value[63] amongst others[64,65], which has previously shown success in other applications such as[29,30]. However, when Thompson sampling is used as the acquisition function, parallel experiments are simultaneously generated as per the batch size. The batch size was determined by the throughput of the experimental system, material availability, and replicate requirements. We note here that the choice of batch size will impact convergence, either by increasing the number of iterations required for convergence (selecting small batch size) or the total number of experiments required to converge (selecting large batch sizes). We also note that the batch sizes could be modified as the iterative learning progresses instead of having a fixed batch size.

## Computational environment and packages

These codes were run on a MacBook Pro with a 2.4 GHz 8-core Intel Core i9 processor, Intel UHD Graphics 630 (1536 MB), and 64 GB 2667 MHz DDR4 memory. As mentioned, GPy, GPyOpt and scipy packages were used for modeling and optimizer implementation, respectively. In addition, NumPy, pandas, and matplotlib, seaborn were used for the data reading, processing, analysis and visualization. No parallelization or Graphical Processing Unit (GPU) capabilities were used in the current implementation.

## Reporting summary

Further information on research design is available in the Nature Portfolio Reporting Summary linked to this article.

## Data availability

The media formulations and the corresponding target data that support the findings of this study and were generated in this study are available in figshare with the identifier(s) https://doi.org/10.6084/m9.figshare.27715134. Source data to create the figures in the paper are provided in Source Data.xlsx file. Source data are provided with this paper.

## Code availability

The codes used to generate the experiments, perform the analyses and generate results in this study is publicly available and has been deposited in GitHub at https://github.com/NHarini-1995/CellCultureBayesianOptimization.git, under MIT license[66]. The specific version of the code associated with this publication is archived in Zenodo and is accessible via https://doi.org/10.5281/zenodo.15466161.

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

## Acknowledgements

We thank the MIT AltHost Research Consortium for the strains of *K. phaffii* used in these studies. This work was funded with support from a SPARC grant from the Broad Institute and the Daniel I.C. Wang (1959) Faculty Research Innovation Fund at MIT. We thank the Mazumdar-Shaw International Oncology fellowship and the Koch Institute at MIT for supporting the research activities of H.N.

## Author contributions

H.N. conceptualized the idea, implemented the algorithm, designed the studies, performed the experiments and analytics for the *K.phaffii* studies, and wrote and revised the manuscript. J.H. designed the protocol and performed experiments and analytics for PBMC experiments. R.B., B.D., L.A.W, and A.A. carried out the PBMC experiments and flow cytometry preparations. Y.Y.T provided lab facilities for the PBMC work and reviewed the manuscript. J.C.L. supervised this work, and wrote and revised the manuscript.

## Competing interests

J.C.L. has interests in Amplifyer Bio, Sunflower Therapeutics PBC, Honeycomb Biotechnologies, OneCyte Biotechnologies, QuantumCyte, and Repligen. J.C.L.'s interests are reviewed and managed under MIT's policies for potential conflicts of interest. The remaining authors declare no competing interests.
