## [Transparent Peer Review file · Nature Communications]

Accelerating Cell Culture Media Development Using Bayesian Optimization-Based Iterative Experimental Design

Corresponding Author: Professor J Christopher Love

Version 0:

Reviewer comments:

Reviewer #1

(Remarks to the Author)

The paper considers applying Bayesian optimization, balancing exploration and exploitation while alternating between experiments, modeling and optimization, to optimize cell culture media. Two cases are considered experimentally, optimization of viability in ex vivo PBMC growth and improving recombinant protein production in yeast. Both cases demonstrate impressive results in terms of the efficient use of experimental data to determine optimal media conditions.

The main novelty of the paper is the application as such, i.e., using the well established and widely used method of Bayesian optimization to optimize cell culture media in two different settings. This is an important contribution in itself, demonstrating a highly efficient use of experiments and data to achieve optimal media compositions. The experimental results are impressive in my view and warrants publication.

While the applications as such are novel, there exists other works proposing a similar approach to cell culture media optimization. For instance, Wang et al. (2024) consider a Bayesian inspired approach to optimize the media in monoclonal antibody production using mammalian cells. However, their work is demonstrated on a simulation model and does not involve experiments.

Having a more theoretical background, I can not comment in any detail on the experimental methodology. However, the theoretical basis for the employed methods appears sound.

Some detailed comments/questions:

121: you state "constrained optimisation problem"; is the constraint referred to here the rather trivial constraint that the ratio of different media must sum to one, or are there other constraints considered in the optimisation?

202: related to the comment above; not clear why the constraint on media blending should limit the accessible region for experiments? Must be due to some specific (restrictive) implementation of the constraint, or?

126: not clear why this second step was performed separately? Would it not be possible to combine the two and then impose the size of subpopulations as a constraint in the optimization? Also, does the second step have no impact whatsoever on the viability?

Ref:

Wang, Y. et al., "Iterative learning robust optimization - with application to medium optimization of CHO cell cultivation in continuous monoclonal antibody production", J Process Control, 1, 137 (2024)

(Remarks on code availability)

Reviewer #2

(Remarks to the Author)

This article presents a method for efficient and cost-effective culture medium optimization. Using this method, the author successfully applied it in two aspects: (1) exploring the optimal commercial medium combinations for maintaining PBMC cell viability and the ratio of active ingredients, and (2) investigating the best carbon source ratio for expressing different foreign proteins in *Pichia pastoris*. The results show that combining Bayesian Optimization (BO) with Gaussian Process (GP) modeling offers significant advantages over traditional methods such as Design of Experiments (DoE) and One-Factor-at-a-Time (OFAT), as it saves experimental costs by achieving better optimization results with fewer trials. If this method could be integrated with automated experimental systems, it would enable more efficient process optimization, greatly enhancing the efficiency of biological process optimization. The work is innovative in its methodological framework, particularly in addressing categorical variables via modified kernel designs and demonstrating resource efficiency (3–30× reduction in experimental burden compared to DoE). The incorporation of transfer learning to extend design spaces and the validation of unexplored regions further highlight the robustness of the approach. The author's use of BO and the exploration of the trade-off between exploitation and exploration to accelerate the optimization process introduces some innovative aspects to the methodology. However, several critical issues limit the current manuscript's impact, and revisions are necessary to strengthen its scientific rigor and translational relevance.

Major concerns :

1. Lack of Comparative Analysis with Alternative ML Methods

The superiority of BO-GP over other machine learning approaches (e.g., random forests, gradient boosting) remains unvalidated. A comparative analysis of prediction accuracy, computational cost, and scalability across methods is essential to justify the choice of GP. In models simulating biological processes, methods such as Random Forest and Bayesian Networks can also achieve similar results to GP. Does GP offer enough advantages over these two methods? Could this be discussed in the paper? Additionally, could methods like XGBoost be used for modeling as well? The author should provide more comparisons and discussions regarding the choice of model in the paper.

2. Insufficient Detail on Hyperparameter Optimization

While the Matern kernel ($\nu=2.5$) and mixed kernel design (Eq. 4) are described, the manuscript omits specifics on hyperparameter tuning (e.g., length scales, noise levels). Clarify whether parameters were optimized via marginal likelihood maximization or cross-validation, and discuss sensitivity to initializations. Matern kernel function with $\nu=3/2$ is typically used for biological experimental optimization rather than $\nu=5/2$, which is generally used in physical simulations. The author should modify this parameter and check whether there are significant differences in the results.

3. Statistical Significance and Noise Quantification

Claims of "no statistically significant improvement" (Line 179) lack supporting p-values or confidence intervals. Include statistical tests (e.g., ANOVA, t-tests) for key comparisons (e.g., Benchmark1 vs. optimized media). Additionally, quantify experimental noise (e.g., biological replicate variability) and its incorporation into the GP model.

4. Biological Mechanism Elucidation

The discussion focuses on algorithmic performance but neglects mechanistic insights. For instance, how do optimized carbon sources (e.g., ribose vs. rhamnose) align with *K. phaffii*'s metabolic pathways? Link findings to known biology (e.g., carbon catabolite repression, secretory stress) to enhance translational relevance.

5. Scalability and Practical Limitations

The computational cost of "2–4 mins/experiment" (Line 342) may hinder large-scale adoption. Provide hardware specifications, parallelization strategies, and benchmarks for higher-dimensional spaces (e.g., >15 factors). Additionally, discuss practical constraints (e.g., media cost, additive stability).

Minor concerns:

1. "process noise was incorporated" (Line 421) is vague. Specify how noise was modeled (e.g., additive Gaussian, heteroscedastic).
2. Revise convoluted sentences (e.g., Lines 254–256) for clarity.
3. Acronyms like "LHS" (Line 425) should be defined at first use.

This work addresses a significant challenge in biological optimization with methodological innovation and practical utility. However, the issues outlined above must be addressed to fully validate the framework's superiority and biological relevance. I recommend major revisions prior to reconsideration for publication. Priority should be given to (1) comparative ML analyses, (2) statistical validation, and (3) mechanistic discussions. With these revisions, the manuscript has the potential to become a benchmark for data-driven media optimization in biotechnology.

(Remarks on code availability)

The code supplies a detailed README, and it seems ok for me to follow the codes

Reviewer #3

(Remarks to the Author)

General Comments:

The authors have developed and tested an algorithm using GP models and Bayesian optimization to speed up media optimization. This is a valuable area of research as these optimization problems often have high dimensionality and noise, thus making them less likely to work well with conventional optimization methods, such as random searches, OFAT, or even DoE. Since these experiments are expensive and time consuming, any kind of acceleration will be worthwhile. It is

noteworthy that the authors tested their algorithm on media for two totally disparate organisms.

Although the authors do not include it in their introduction, this is not a new field, even for media optimization. As the authors point out, other researchers have used many types of modeling to accelerate process optimization. And, as the authors also point out, many of these applications of modeling have been mostly trivial, improving models with neural networks or other flexible, data driven modeling methods, but not using the more advanced models to more efficiently accumulate knowledge (and choose new experiments). However, there has been some work in this field including work from many years ago such as Zhang et al. (reference 7 below) and others. More recently, there have been papers that have been published that address nearly identical methods as the current work but go further some aspects of their methodology than this manuscript (specifically, references 3 and 4 below; but also 1, 5, and 6). In some of these papers, they also run extremely efficient DoE experimentally along side the newer methods and actually compare the methods. Many of these newer methods reduce experimentation by about 30-40% compared with good DoE methods. The larger reduction noted in the current manuscript is likely because the DoE control number of experiments is calculated incorrectly (as noted below) or because the correct kind of DoE approach is not chosen for comparison or both. Aside from clarifying methods as noted below, this manuscript would benefit from stressing parts of the algorithm that are unique. In most of the studies noted above, categorical variables were not used, so this can be emphasized. In addition, the transfer knowledge component of this manuscript is interesting, though it is not clear that it gives a better optimum than starting over. That is, would starting from scratch have resulted in a better optimum given that biases would not have to be overcome by starting with the existing model. The dimensionality of the experimentation used as examples is also quite small, actually lending itself in some cases, to well-designed DoE.

Specific Comments:

Abstract:

The abstract should be specific on how this work furthers already published work. The comparison to standard DoE should make it clear that these are estimates and not an actual comparison.

Introduction:

Lines 37-38. Rephrase the list of components. As an example, nutrients, amino acids, and nitrogen sources could all be the same thing. You might have meant to say something like nutrients such as carbon sources, nitrogen sources, minerals, growth hormones, etc.

Lines 47-48. There is a statement referring to bacterial cultivation and fermentation, yet not all of your references for this statement refer to microbial growth.

Line 51. This sentence ("These studies, however,...") ignores all of the previous work in this area related to optimizing media. A short list of references in this field is given below. A few sentences could be used to summarize this work, so the reader does not think that this area of research is brand new with no previous published literature. You could also describe the limitations of this previous work to justify your own approach.

Lines 78-81. Include a description of the previous work using BO for media optimization (references below, especially 3 and 4, Cosenza et al., 2023 and Cosenza et al., 2022, respectively) and what limitations of this work are addressed by the current work in this manuscript. In some cases, the previous methods are significantly more sophisticated than the one used here, though there are differences.

Line 84. Include the number of factors in this optimization. In addition, somewhere in the introduction, you should include a description of the expected dimensionality of media optimization problems. This should help you justify the BO approach since smaller dimensionality is often well handled by DoE approaches.

Results:

Lines 118-120. Rephrase. You are not hypothesizing that different formulations will have different nutrients, just that different nutrients will give different results.

Lines 124-125. You only have three independent variables, so a full response surface (CCD) would be 8 factorial points + 6 axial points + 1 centerpoint. Would this be more efficient? How many experiments are in the initial batch? Are these included in the 24? Be clearer on the 80%--just use the numbers--viability increased from blank percent to blank percent.

Line 128. "To address this imbalance,...", others have used various approaches to multi-objective optimization, including the idea of "desirability" and others. Is there a reason not to use multi-objective optimization in this case study to cover cell growth and differentiation?

Lines 131-132. People don't use full factorial designs in practice, except with very small numbers of factors. It is more common to use fractional factorial designs to explore factor ranges (sometimes using lower resolution designs) and then zero in using central composite designs or response surface methodology. It would be good to account for this in your comparison, especially if you aren't verifying the DoE number of experiments experimentally. There are some examples of DoE approaches in your references and in the references below (for example, reference 8 below). Also, for factorial design, it would be 28 experiments for full factorial, not 108 as you have listed here and in your figure.

Line 136. "With as few as 12..." Is this using the optimized media? Is this 12 additional experiments after an initial database is generated for the nine new factors or just 12 experiments? How does this compare to something like a Plackett-Burman Design in terms of number of experiments? Are these in one batch or several batches of smaller numbers of experiments? Expand here or refer to methods.

Line 143. For the PBMC media, would you have ended up with a better media if you had added the growth factors to only 1-3 of the base media instead of four? You can comment on this in the discussion.

Line 148. It might be good at the first reference of *K. phaffii* to let readers know that this organism was formerly known as *P. pastoris*. I think more readers interested in your work will recognize this as an important industrial host.

Line 151. "Induce production" I would be specific here and say production of recombinant protein, as that is what you are optimizing here. Production could also mean biomass.

Line 152. Reference 42 is not an actual reference. Please add one.

Line 162. Table 1 is confusing in its current format. Reformat to make the Benchmarks and possible carbohydrate additions more clear. The C additions look like they are already paired up as possibilities which I think is not your intent.

Lines 179-181. Discuss what these media are and how different they are from the benchmarks. Are the carbon sources different? How many are chosen or is this constrained?

Lines 188-191. Not sure where these numbers come from and what a comprehensive search means. Please clarify for each of the optimization methods, what is the optimization method and how were the numbers calculated.

Lines 196-198. Did you test other initial designs? It may not make that much difference, but it seems like it could. How about the number of experiments in the initial design? Will this impact convergence of your algorithm? What is the trade off?

Lines 208-234. These results seem to match what others have found. I'm not sure how novel this is, but it is definitely interesting. Could each of these figures be placed in the sections with their respective optimizations? This might make more sense than going back and forth between case studies.

Lines 244-248. Can you assure that the algorithm found the best combination or just a good combination? It didn't exhaustively search the space. What if you went further? Where is the trade-off? You could comment on this in the discussion. You seem to be basing your end point on the model predictions matching the experiment. This makes sense, but how do you know that your model doesn't match experiment in an unexplored region that happens to have a better optimum?

Line 260. "to the first case." Do you mean second case?

Lines 268-274. This seems like a unique idea that could be very helpful as cases change for optimization. Do you think you will reach the same optimum?

Lines 283-285. Your comparison seems unlikely, as there are definitely DoE methods for looking through 12 variables in less than 200 experiments.

Discussion:

Lines 308-311. Again, efficient DoE methods will take a lot fewer experiments than what you have listed in your figures. While this makes your method look really good compared to DoE, it's unfortunate because people familiar with DoE may discount your work. Your method is likely better than really efficient DoE methods. Because you haven't tried one of these experimentally for comparison, you could estimate differently or refer to previous work that actually did these comparisons.

Lines 316-320. You haven't proven that the transfer learning is better than starting over or larger from beginning. You are simply showing that it works ok.

Lines 331-332. There is active feedback from in DoE from iteration to iteration (and there will almost always be iterations)--just not an accumulation of knowledge, making it less efficient.

Lines 335-336. This assumes that the ranges don't move in DoE. This would be a bad DoE approach that people wouldn't normally use. The advantage is more that DoE is a gradient optimization method and is more likely to get stuck in local minima than your method.

Methods:

Lines 376-380. These two sentences are likely copied from a protocol. Reword for the paper.

Lines 380-382. Clarify how you combined replicates. This sentence is not clear and not consistent with subsequent phrasing

later in the manuscript on Lines 385-386.

Lines 410-412. Reword to make sense.

Line 414. Change “outgrown” to “grown.”

Lines 420-421. Explain how duplicates give enough data to describe variability. Also, it is not clear how you use control experiments across batches to assess variability. Please clarify.

Lines 425-426. This is not a sentence. Please edit.

Figures and Tables:

Lines 654-657. What is iteration 0 and iteration 1 in Figure 2d? Figure 2c numbers need to be corrected as per note above.

Figure 3c. The rows of this chart are not clear. Please modify. Again figure 3c numbers seem to be off—or at least need better explanation.

Figure 4b is difficult to interpret. Could this be shown in one 3D graph? Figure 4c would be easier to interpret with different colors or patterns, as two of the colors look identical in gray scale.

Figure 5. The exploration and exploitation are difficult to see in these figures. Is there a way to show how the values of the different factors change over time instead?

References for Review:

1. Claes, E., Heck, T., Coddens, K., Sonnaert, M., Schrooten, J., & Verwaeren, J. (2024). Bayesian cell therapy process optimization. *Biotechnology and Bioengineering*, 121, 1569–1582. <https://doi.org/10.1002/bit.28669>
2. Coleman, M.C. and Block, D.E. (2007), Nonlinear experimental design using Bayesian regularized neural networks. *AIChE J.*, 53: 1496-1509. <https://doi.org/10.1002/aic.11175>
3. Cosenza Z, Block DE, Baar K, Chen X. Multi-objective Bayesian algorithm automatically discovers low-cost high-growth serum-free media for cellular agriculture application. *Eng Life Sci.* 2023; 23:e2300005. <https://doi.org/10.1002/elsc.202300005>
4. Cosenza, Z., Astudillo, R., Frazier, P. I., Baar, K., & Block, D. E. (2022). Multi-information source Bayesian optimization of culture media for cellular agriculture. *Biotechnology and Bioengineering*, 119, 2447–2458. <https://doi.org/10.1002/bit.28132>
5. K. Watanabe, T.-Y. Chiou, M. Konishi, Optimization of medium components for protein production by *Escherichia coli* with a high-throughput pipeline that uses a deep neural network, *Journal of Bioscience and Bioengineering*, Volume 137, Issue 4, 2024, Pages 304-312, ISSN 1389-1723 <https://doi.org/10.1016/j.jbiosc.2024.01.005>.
6. Xiao, W., Shi, X., Huang, H., Wang, X., Liang, W., Xu, J., Liu, F., Zhang, X., Xu, G., Shi, J., & Xu, Z. (2024). Enhanced synthesis of S-adenosyl-L-methionine through combinatorial metabolic engineering and Bayesian optimization in *Saccharomyces cerevisiae*. *Biotechnology Journal*, 19, e2300650. <https://doi.org/10.1002/biot.202300650>
7. Zhang, G. and Block, D.E. (2009), Using highly efficient nonlinear experimental design methods for optimization of *Lactococcus lactis* fermentation in chemically defined media. *Biotechnol Progress*, 25: 1587-1597. <https://doi.org/10.1002/btpr.277>
8. Zhang G, Mills DA, Block DE. Development of chemically defined media supporting high-cell-density growth of lactococci, enterococci, and streptococci. *Appl Environ Microbiol.* 2009 Feb;75(4):1080-7. doi: 10.1128/AEM.01416-08. Epub 2008 Dec 12. PMID: 19074601; PMCID: PMC2643557.

(Remarks on code availability)

Version 1:

Reviewer comments:

Reviewer #1

(Remarks to the Author)

I find the response to my comments largely satisfactory, both in the response letter as well as in terms of corresponding modifications of the paper. However, I find it very peculiar that they do not respond in any way to my comment on a recently published work proposing a very similar methodology to essentially the same application, that is, optimizing cell culture media composition using a Bayesian like approach (Wang et al., 2024, full reference in my original review). Although Wang

et al. only demonstrated the method on a simulation model, it is highly relevant to the current work and should hence be commented upon and referenced.

(Remarks on code availability)

Reviewer #2

(Remarks to the Author)

All my concerns have been addressed, and the revised manuscript is improved a lot. I recommend this work to be published!

(Remarks on code availability)

Point-By-Point Response

Reviewer #1 (Remarks to the Author):

The paper considers applying Bayesian optimization, balancing exploration and exploitation while alternating between experiments, modeling and optimization, to optimize cell culture media. Two cases are considered experimentally, optimization of viability in ex vivo PBMC growth and improving recombinant protein production in yeast. Both cases demonstrate impressive results in terms of the efficient use of experimental data to determine optimal media conditions.

The main novelty of the paper is the application as such, i.e., using the well established and widely used method of Bayesian optimization to optimize cell culture media in two different settings. This is an important contribution in itself, demonstrating a highly efficient use of experiments and data to achieve optimal media compositions. The experimental results are impressive in my view and warrants publication.

While the applications as such are novel, there exists other works proposing a similar approach to cell culture media optimization. For instance, Wang et al. (2024) consider a Bayesian inspired approach to optimize the media in monoclonal antibody production using mammalian cells. However, their work is demonstrated on a simulation model and does not involve experiments.

Having a more theoretical background, I can not comment in any detail on the experimental methodology. However, the theoretical basis for the employed methods appears sound.

We thank the reviewer for their positive remarks on the importance of the research, theoretical soundness and the validation of the approach on two very different experimental systems.

Some detailed comments/questions:

121: you state "constrained optimisation problem"; is the constraint referred to here the rather trivial constraint that the ratio of different media must sum to one, or are there other constraints considered in the optimisation?

Yes, that is correct, the constraint is a linear-equality constraint enforcing the sum of media composition to be equal to 1.

Revised Text (Pg. 7, Ln 154-157): *"This framing yields a constrained optimization problem of continuous design factors with a linear equality constraint, such that the relative contributions of the different media in the blend sum to 100%"*

202: related to the comment above; not clear why the constraint on media blending should limit the accessible region for experiments? Must be due to some specific (restrictive) implementation of the constraint, or?

We apologize for the confusion regarding the constraint. We intended that in an unconstrained optimization setting, the accessible design space spans the entire hypercube defined by the bounds. In contrast, for constrained optimization (e.g., media blending), the accessible space is the portion of the hypercube where the constraints are satisfied. For clarity, we have reworded this sentence in the revised manuscript.

Revised Text (Pg 10, Ln 263-265): *“We note that the feasible region to plan experiments in this example is a non-cuboidal design space defined by the constraint imposed on media blending. That is, the sum of the ratios of different media type should equal one (Black dashed line, Fig. 4B).”*

126: not clear why this second step was performed separately? Would it not be possible to combine the two and then impose the size of subpopulations as a constraint in the optimization?

We agree that the optimization could be approached in alternative ways. If viewed solely from the perspective of an optimization exercise, such a formulation of the problem might result in an improved optimum when considering both the basal media blend together with the cytokine composition.

Our rationale for splitting this task into two sequential optimizations was driven by the staging of the application. An optimized basal nutrient media could provide a basis for several related biological applications requiring lymphocytic cell populations including culturing hematopoietic cancer cells, CART cells, and hematopoietic toxicity, among others. The sequential approach here allows optimizing the cytokine/chemokine composition to modulate required task-specific properties. This approach would also allow emulation of the nutrient and signaling environment in vivo that could be used for future studies to draw mechanistic conclusions.

We have added a sentence to summarize this point in the revised manuscript.

Revised Text (Pg 6-7, Ln 143-151): *“Alternatively, both the basal media and mixture of cytokines used could be jointly optimized, potentially leading to an improved formulation. This approach, however, introduces a trade-off in iterative optimizations. By splitting the task into two sequential optimizations, the determined basal nutrient media can serve as a basis for related specific applications involving lymphocytic cell populations (e.g., culturing hematopoietic cancer cells, CAR T cells etc). In this way, only additional optimization of the cytokine/chemokine composition is necessary to modulate the subsequent required properties. This approach could also allow emulation of the nutrient and signaling environment in vivo in studies to assess the underlying biological mechanisms involved.”*

- Also, does the second step have no impact whatsoever on the viability?

The second step can affect viability and the second optimization, in fact, accounts for the viability in the objective calculation. This point has been clarified in the revised manuscript in the methods section.

Revised Text (Pg 21 Ln 586-589): *“To determine the PBMC basal media blend, cell viability was used as the target property whereas for the cytokine optimization an aggregated objective function was used to consider both cell viability and cell differentiation (Eq. 7). Finally for the case study with K.phaffii, specific productivity was used as the target.”*

Ref:

Wang, Y. et al., "Iterative learning robust optimization - with application to medium optimization of CHO cell cultivation in continuous monoclonal antibody production", J Process Control, 1, 137 (2024)

Reviewer #2 (Remarks to the Author):

This article presents a method for efficient and cost-effective culture medium optimization. Using this method, the author successfully applied it in two aspects: (1) exploring the optimal commercial medium combinations for maintaining PBMC cell viability and the ratio of active ingredients, and (2) investigating the best carbon source ratio for expressing different foreign proteins in *Pichia pastoris*. The results show that combining Bayesian Optimization (BO) with Gaussian Process (GP) modeling offers significant advantages over traditional methods such as Design of Experiments (DoE) and One-Factor-at-a-Time (OFAT), as it saves experimental costs by achieving better optimization results with fewer trials. If this method could be integrated with automated experimental systems, it would enable more efficient process optimization, greatly enhancing the efficiency of biological process optimization. The work is innovative in its methodological framework, particularly in addressing categorical variables via modified kernel designs and demonstrating resource efficiency (3–30× reduction in experimental burden compared to DoE). The incorporation of transfer learning to extend design spaces and the validation of unexplored regions further highlight the robustness of the approach. The author’s use of BO and the exploration of the trade-off between exploitation and exploration to accelerate the optimization process introduces some innovative aspects to the methodology. However, several critical issues limit the current manuscript’s impact, and revisions are necessary to strengthen its scientific rigor and translational relevance.

Major concerns :

1. Lack of Comparative Analysis with Alternative ML Methods

The superiority of BO-GP over other machine learning approaches (e.g., random forests, gradient boosting) remains unvalidated. A comparative analysis of prediction accuracy, computational cost, and scalability across methods is essential to justify the choice of GP. In models simulating biological processes, methods such as Random

Forest and Bayesian Networks can also achieve similar results to GP. Does GP offer enough advantages over these two methods? Could this be discussed in the paper? Additionally, could methods like XGBoost be used for modeling as well? The author should provide more comparisons and discussions regarding the choice of model in the paper.

We appreciate the point raised by the reviewer on the relative merits of the Gaussian Process models with Bayesian Optimization compared to other ML approaches.

We have added a discussion in the introduction explaining the rationale for the choice of Gaussian Processes for the applications addressed in this study.

Revised Text (Pg 4-5, Ln 78-92): *“GPs are suitable for unbiased learning of smooth response functions compared to alternative ML algorithms, such as tree-based models that are bound by splitting rules and learn discontinuous or piecewise continuous decision boundaries. Furthermore, GPs can include prior beliefs about the system, incorporate process noise in its implementation, and obtain confidence in its predictions by associating higher uncertainty with unexplored parts of the design space¹⁸. Most other classical ML models don’t allow the explicit incorporation of prior assumptions or process noise and inherently provide point estimates¹⁹. While uncertainty can be estimated using ensembling techniques, these estimates do not directly correspond to the positions of the data points within the design space. These abilities are important for intrinsically noisy biological systems that require expensive experimentation. In this context, approaches that can encode prior beliefs could reduce the overall experimental burden of optimization. Furthermore, GPs are efficient for handling small volumes of data (common with biological systems) compared to alternative tree-based approaches, which often perform well with larger volumes of data (e.g. a pre-existing database is available)¹⁷. Additionally, custom kernels can be designed for GP models to suit the specific needs of an application.”*

2. Insufficient Detail on Hyperparameter Optimization

While the Matern kernel ($\nu=2.5$) and mixed kernel design (Eq. 4) are described, the manuscript omits specifics on hyperparameter tuning (e.g., length scales, noise levels). Clarify whether parameters were optimized via marginal likelihood maximization or cross-validation, and discuss sensitivity to initializations.

The hyperparameters were determined using marginal likelihood maximization

Revised Text (Pg 21, Ln 569-571): *“The length scale in the continuous kernel and the trade-off parameter (α) in the mixture kernel is a hyperparameter that is updated by maximizing the marginal likelihood based on data, refining the priors and resulting in the posterior distribution.”*

The length scales were initialized to the default settings in GPy and the trade-off parameter (where relevant) was initialized to 0.5, giving equal weight to both

additive and multiplicative terms at the beginning. In every iteration, the training errors were monitored through root mean squared error (RMSE) to ensure the success of parameter optimization. Using *K.phaffii* as the example, the stability of the approach to initialization was verified a-posteriori (Supplementary information, Fig S3). We have revised the manuscript to include this description in the manuscript.

Revised Text (Pg 21, Ln 572-576): *“The length scales were initialized to the default settings in GPy and the trade-off parameter (where relevant) was initialized to 0.5 giving equal weight to both additive and multiplicative terms at the beginning. With each iteration, the training errors were monitored through RMSE in training to ensure the success of parameter optimization. Using *K.phaffii* as an example, we verified the stability of this approach to initialization a-posteriori (Supplementary information, Fig. S3).”*

- Matern kernel function with $\nu=3/2$ is typically used for biological experimental optimization rather than $\nu=5/2$, which is generally used in physical simulations. The author should modify this parameter and check whether there are significant differences in the results.

We agree that the value used is important. We have revised the manuscript to clarify the details of the implementation. Matern with $\nu=3/2$ was used for the higher noise system (the PBMC case study) while Matern with $\nu=5/2$ was used for the *K.phaffii* studies since the categorical overlap kernel could be used only with a smooth kernel (at least twice differential).

An a-posteriori validation indicated that the Matern kernel with $\nu=5/2$ gives similar predictive performance to the kernel with $\nu=3/2$ for the PBMC study as well.

All these data are now included in the supplementary information (Fig. S2) and we have added the relevant supporting text to the revised manuscript.

Revised text (Pg 20, Ln 550-556): *“Common values of ν for this kernel are 1.5 and 2.5. The first value yields once differentiable functions and the second results in twice differentiable functions. The categorical overlap kernel (Eq. 3) is compatible with at least twice differentiable kernels such as Radial Basis Function (RBF) or Matern ($\nu = 2.5$)⁵⁷, motivating the selection of the kernel. For the PBMC case study (considering only continuous design factors) with higher noise, a Matern kernel with $\nu = 1.5$ was used. (We note that both $\nu = 1.5$ and 2.5 performed similarly, however, for these data (Supplementary Information, Fig. S2)).”*

3. Statistical Significance and Noise Quantification

Claims of "no statistically significant improvement" (Line 179) lack supporting p-values

or confidence intervals. Include statistical tests (e.g., ANOVA, t-tests) for key comparisons (e.g., Benchmark1 vs. optimized media).

We thank the reviewer for pointing out this oversight. We have added the p-values for the key comparisons in Figure 3 and revised the text to reflect this change.

Revised text (Pg 9, 218-220): *“Finally, for trastuzumab, we observed a 3-fold improvement compared to Benchmark2 (6 mg/L/OD600 vs 2 mg/L/OD600) and no statistically significant improvement compared to Benchmark1 ($p = 0.09$)”*

- Additionally, quantify experimental noise (e.g., biological replicate variability) and its incorporation into the GP model.

An additive Gaussian noise was used to account for the process noise, incorporated through the White Noise kernel. The variability of biological replicates was used to calculate the process noise used in the model. This point has now been clarified in the revised manuscript.

Revised text (Pg 19, Ln 514-521): *“Biological duplicates were run for all designed experiments and the average value of the biological duplicate was considered for the modeling. Cultivation using media indicated in Benchmark 1 was run as a control experiment on each iteration. The process noise was calculated as the variance across the control experiment for all instances including the contemporaneous iteration. For building the initial model, the variance was calculated from a preliminary experimental campaign to select molecules expressed with Benchmark 1 media. This estimated process noise was incorporated into the modeling (see Gaussian Processes).”*

Revised text (Pg 21, Ln 566-568): *“To this kernel, process noise was added through a white noise kernel with a fixed variance computed based on the variation in the replicates of the control experiment for each iteration/across different iterations.”*

4. Biological Mechanism Elucidation

The discussion focuses on algorithmic performance but neglects mechanistic insights. For instance, how do optimized carbon sources (e.g., ribose vs. rhamnose) align with *K. phaffii*'s metabolic pathways? Link findings to known biology (e.g., carbon catabolite repression, secretory stress) to enhance translational relevance.

We agree the findings are interesting regarding the preferences for carbon sources. These alternative co-feeds are not well studied in the literature, especially the impact of these on recombinant protein production quantities/qualities. We have added a summary of current insights related to biomass/growth behaviors and discussed approaches for future work that could yield further mechanistic insights.

Revised Text (Pg 9-10, Ln 226-238): *“The alternative carbon sources considered in this work have not been widely studied for their impact on recombinant protein production in *K.phaffii*. Sorbitol has been used as a co-fed carbon source for generating biomass with *K.phaffii*⁵¹. L-rhamnose, another carbon source metabolized by *K.phaffii*⁵², has shown improved production for HSA here, suggesting a potential alternative to sorbitol. Other carbon sources, such as glycerol, glucose, fructose, and mannose, are known to support growth/biomass accumulation while having a repressive impact on the *pAOX1* promoter^{51,52}, making them unsuitable candidates to promote protein production. Many carbon sources not known to be metabolized by *K.phaffii*, including D-arabinose, D-ribose, D-xylose, galactose, lactose, xylitol and sucrose⁵² interestingly showed benefits as a co-feed to enhance recombinant protein production here. Which carbon sources were beneficial, however, depended on the protein produced. How these different carbon sources influence protein production would merit further investigations to assess their influence on the cellular states.”*

5. Scalability and Practical Limitations

The computational cost of "2–4 mins/experiment" (Line 342) may hinder large-scale adoption.

The temporal cost of computation of 2-4 mins/experiment translates to ~40 -50 mins for a batch of 12 experiments, which is reasonable for lab scale implementations ‘as is’. We note that this cost was only manifest when considering categorical design spaces with the constant-liar approach for batching the experiments. Additionally, in this work, we did not optimize for computational efficiency and did not perform parallelization, or use GPU capacities. Both of these well-defined methods could reduce the computational time significantly. We have added a sentence to highlight this point in the revised manuscript.

Revised Text (Pg 16, Ln 425-429): *“Additionally, the current implementation does not account for parallelization or utilize GPU acceleration, both of which could significantly enhance computational efficiency. Further advances to enhance the computational efficiency using Sparse Gaussian Process Regression could also reduce the memory required, and enable efficient processing of larger numbers of factors and sizes of datasets⁵⁰.”*

- Provide hardware specifications, parallelization strategies, and benchmarks for higher-dimensional spaces (e.g., >15 factors).

We have added the hardware specification under a newly created subsection in the Methods section of the revised manuscript: “Computational Environment and Packages” [Line 618-623].

As noted above, no parallelization or GPU computing was applied in the current implementation but is a direction we will consider for future work. We have also highlighted this point in the revised manuscript [Line 425-429].

- Additionally, discuss practical constraints (e.g., media cost, additive stability).

We have revised the manuscript to acknowledge that considering the cost penalty, ability to make stable stock solutions, and environmental impact of these components will all impact the practical choice and scalability of the final media for the application. We have acknowledged these additional constraints in the Discussions section.

Revised Text (Pg 16, Ln 431-436): *“The scalability of the methods here, however, also depends on practical considerations in media optimization, such as the cost, solubility, stability/cross-compatibility, and environmental impact of the components. These considerations could be included in the optimization as additional objectives and (or) constraints and addressed through further advances of the current framework.”*

Minor concerns:

1. “process noise was incorporated” (Line 421) is vague. Specify how noise was modeled (e.g., additive Gaussian, heteroscedastic).

We have revised the text to clarify this statement in the model/kernel description [Ln 566-568].

2. Revise convoluted sentences (e.g., Lines 254–256) for clarity.

We rephrased this sentence for clarity [Ln 329-332].

3. Acronyms like “LHS” (Line 425) should be defined at first use.

We have defined the acronyms on first use as suggested in the revised manuscript [Ln 523].

This work addresses a significant challenge in biological optimization with methodological innovation and practical utility. However, the issues outlined above must be addressed to fully validate the framework’s superiority and biological relevance. I recommend major revisions prior to reconsideration for publication. Priority should be given to (1) comparative ML analyses, (2) statistical validation, and (3) mechanistic discussions. With these revisions, the manuscript has the potential to become a benchmark for data-driven media optimization in biotechnology.

We appreciate the reviewer’s recognition of the importance and innovation of this work, and the suggestions to improve the clarity and presentation of the

manuscript that were provided.

Reviewer #2 (Remarks on code availability):

The code supplies a detailed README, and it seems ok for me to follow the codes

Reviewer #3 (Remarks to the Author):

General Comments:

The authors have developed and tested an algorithm using GP models and Bayesian optimization to speed up media optimization. This is a valuable area of research as these optimization problems often have high dimensionality and noise, thus making them less likely to work well with conventional optimization methods, such as random searches, OFAT, or even DoE. Since these experiments are expensive and time consuming, any kind of acceleration will be worthwhile. It is noteworthy that the authors tested their algorithm on media for two totally disparate organisms.

We thank the reviewer for their positive remarks on the importance of the research, and the effort required to validate the approach on two very different experimental systems.

Although the authors do not include it in their introduction, this is not a new field, even for media optimization. As the authors point out, other researchers have used many types of modeling to accelerate process optimization. And, as the authors also point out, many of these applications of modeling have been mostly trivial, improving models with neural networks or other flexible, data driven modeling methods, but not using the more advanced models to more efficiently accumulate knowledge (and choose new experiments). However, there has been some work in this field including work from many years ago such as Zhang et al. (reference 7 below) and others. More recently, there have been papers that have been published that address nearly identical methods as the current work but go further some aspects of their methodology than this manuscript (specifically, references 3 and 4 below; but also 1, 5, and 6). In some of these papers, they also run extremely efficient DoE experimentally along side the newer methods and actually compare the methods. Many of these newer methods reduce experimentation by about 30-40% compared with good DoE methods. The larger reduction noted in the current manuscript is likely because the DoE control number of experiments is calculated incorrectly (as noted below) or because the correct kind of DoE approach is not chosen for comparison or both. Aside from clarifying methods as noted below, this manuscript would benefit from stressing parts of the algorithm that are unique. In most of the studies noted above, categorical variables were not used, so this can be emphasized. In addition, the transfer knowledge component of this manuscript is interesting, though it is not clear that it gives a better optimum than starting over. That is, would starting from scratch have resulted in a better optimum given that biases would not have to be overcome by starting with the existing model. The dimensionality

of the experimentation used as examples is also quite small, actually lending itself in some cases, to well-designed DoE.

We have addressed the specific suggestions/concerns raised by the reviewer in their detailed comments below.

Specific Comments:

Abstract:

The abstract should be specific on how this work furthers already published work. The comparison to standard DoE should make it clear that these are estimates and not an actual comparison.

We have modified the sentence to reflect this point and make it clear in the revised manuscript.

Revised Text (Pg 2, Ln 22-24): *“We identified conditions with improved outcomes for both applications compared to the initial standard media using 3 to 30 times fewer experiments than that estimated for other methods such as the standard Design of Experiments.”*

Introduction:

Lines 37-38. Rephrase the list of components. As an example, nutrients, amino acids, and nitrogen sources could all be the same thing. You might have meant to say something like nutrients such as carbon sources, nitrogen sources, minerals, growth hormones, etc.

We thank the reviewer for pointing out this inaccuracy. We have revised manuscript to rephrase the list of components [Ln 37-39].

Lines 47-48. There is a statement referring to bacterial cultivation and fermentation, yet not all of your references for this statement refer to microbial growth.

We appreciate the reviewer highlighting this oversight. We have corrected the citation in this statement [Ln 51].

Line 51. This sentence (“These studies, however,...) ignores all of the previous work in this area related to optimizing media. A short list of references in this field is given below. A few sentences could be used to summarize this work, so the reader does not think that this area of research is brand new with no previous published literature. You could also describe the limitations of this previous work to justify your own approach.

Lines 78-81. Include a description of the previous work using BO for media optimization (references below, especially 3 and 4, Cosenza et al., 2023 and Cosenza et al., 2022, respectively) and what limitations of this work are addressed by the current work in this

manuscript. In some cases, the previous methods are significantly more sophisticated than the one used here, though there are differences.

We appreciate the reviewer highlighting these other studies. We have revised the manuscript to address the contributions our studies make that are new for the application considered here (media optimization).

Revised Text (Pg 5, Ln 102-104):” *Some studies have also demonstrated these approaches for designing and optimizing cell culture media considering multiple objectives and information sources. These, however, use unconstrained–continuous design factors in their optimizations. Here, we demonstrate the application of a BO-based framework to efficiently optimize the composition of cell culture media considering complex design spaces that include both constraints and categorical variables.*”

Line 84. Include the number of factors in this optimization. In addition, somewhere in the introduction, you should include a description of the expected dimensionality of media optimization problems. This should help you justify the BO approach since smaller dimensionality is often well handled by DoE approaches.

We have added the expected dimensionality of media optimization now in the introduction [Ln 39-41]. We agree that smaller design spaces may be handled well with DoE, at least from the perspective of resources required. We posit, however, that the biases and underlying assumptions imposed by DoE might result in missing optimal solutions if the real system has non-linear interactions/response surfaces (as many biological systems do).

Furthermore, in this work, we considered categorical variables with 19 levels (when considering the smaller design space of four design factors). This added complexity scales the problem quickly since (i) DoEs are not designed to plan experiments in qualitative spaces and (ii) possible modifications would require either considering them as binary variables or replicating continuous factors in the design across all categories. Both these approaches increase the number of experiments required by standard DoEs.

To highlight these aspects, we have revised the manuscript to include a brief discussion in the introduction section [Ln 65-74]. We have also provided our rationale for calculating the number of experiments in the different scenarios in the supplementary information (Table S1, S2, S3) and appropriately cross-referenced in the main text [Ln 249-250, Ln 165-167, Ln 243-245]. Finally, we have augmented the calculations for the number of experiments to include additional DoE approaches based on the suggestions of the reviewer [Figure 2C, Figure 3D].

Results:

Lines 118-120. Rephrase. You are not hypothesizing that different formulations will

have different nutrients, just that different nutrients will give different results.

We had implied here that the different commercial media likely have a different selection and composition of nutrients and blending them would result in unique formulation with the potential to improve cell health/survival. We have revised the sentence to improve the clarity.

Revised Text (Pg 7, Ln 152-154): *“The different commercial formulations of media comprise different sets and (or) quantities of nutrients, hormones, and growth factors. We hypothesized that combining these in different ratios could yield a new composition capable of maintaining high cell viability (>70%).”*

Lines 124-125. You only have three independent variables, so a full response surface (CCD) would be 8 factorial points + 6 axial points +1 centerpoint. Would this be more efficient?

We agree with the reviewer that there are only 3 independent variables. The 4 design factors, however, are bound by an equality linear constraint. To the best of our knowledge, the classical CCD (whose number of experiments calculation is stated above) works for cuboidal design spaces and is not suitable for such constrained design spaces. The classical CCD considers two levels, -1 and 1, corresponding to 0% and 100% in this case, for each independent design factor. The factorial points generated for this design as per the approach suggested by the reviewer will be as follows:

-1	-1	-1
+1	-1	-1
-1	+1	-1
+1	+1	-1
-1	-1	+1
+1	-1	+1
-1	+1	+1
+1	+1	+1

The points marked in red are not feasible under the current constraints of our optimization problem. So, though the number might be smaller, unfortunately this approach may not be suitable/efficient for this problem.

- How many experiments are in the initial batch? Are these included in the 24?

The 24 experiments are the total number of experiments including the initial batch. We have clarified this point in the revised manuscript.

Revised Text (Pg 7, Ln 158-160): *“With 24 total experiments (split into batches of 6 experiments over four iterations, 0 to 3), an optimized blend of the media showed a statistically improved viability of 75-80% compared to individual media which resulted in viability of 60% (Fig. 2B).”*

- Be clearer on the 80%--just use the numbers--viability increased from blank percent to blank percent.

As noted above, the 80% is the absolute viability and not a calculated entity with respect to a different condition. We have rephrased this sentence in the revised manuscript [Ln 158-160].

Line 128. “To address this imbalance,...”, others have used various approaches to multi-objective optimization, including the idea of “desirability” and others. Is there a reason not to use multi-objective optimization in this case study to cover cell growth and differentiation?

We agree that the optimization could be approached in alternative ways. If viewed solely from the perspective of an optimization exercise, such a formulation of the problem might result in an improved optimum when considering both the basal media blend together with the cytokine composition.

Our rationale for splitting this task into two sequential optimizations was driven by the staging of the application. An optimized basal nutrient media could provide a basis for several related biological applications requiring lymphocytic cell populations including culturing hematopoietic cancer cells, CART cells, and hematopoietic toxicity, among others. The sequential approach here allows optimizing the cytokine/chemokine composition to modulate required task-specific properties. This approach would also allow emulation of the nutrient and signaling environment in vivo that could be used for future studies to draw mechanistic conclusions.

We have added a sentence to summarize this point in the revised manuscript.

Revised Text (Pg 6-7, Ln 143-151): *“Alternatively, both the basal media and mixture of cytokines used could be jointly optimized, potentially leading to an improved formulation. This approach, however, introduces a trade-off in iterative optimizations. By splitting the task into two sequential optimizations, the determined basal nutrient media can serve as a basis for related specific applications involving lymphocytic cell populations (e.g., culturing hematopoietic cancer cells, CAR T cells etc). In this way, only additional optimization of the cytokine/chemokine composition is necessary to modulate the subsequent required properties. This approach could also allow emulation of the nutrient and signaling environment in vivo in studies to assess the underlying biological mechanisms involved.”*

Lines 131-132. People don't use full factorial designs in practice, except with very small numbers of factors. It is more common to use fractional factorial designs to explore factor ranges (sometimes using lower resolution designs) and then zero in using central composite designs or response surface methodology. It would be good to account for this in your comparison, especially if you aren't verifying the DoE number of

experiments experimentally. There are some examples of DoE approaches in your references and in the references below (for example, reference 8 below). Also, for factorial design, it would be 28 experiments for full factorial, not 108, as you have listed here and in your figure.

We would like to highlight that the Full factorial numbers are reported under the “Factorial” label in the original figure. The number reported under “Full Screen” is for a exhaustive/grid search approach, which was highlighted as is infeasible, now re-emphasized also in the revised manuscript.

Bayesian Optimization (in contrast to the statistical DoE approaches) considers the entire continuum of the design space instead of discretizing the space. Thus, an exhaustive search with a finer grid (here approximated with 10 levels) is added as a comparison to demonstrate the spread of the design space. We have also provided this rationale for the choice of this comparison in the revised manuscript.

Revised Text (Pg 10, Ln 243-250): *“This total experimental load was ~ 2.5 – 3 times lower than the predicted number of experiments required for standard designs of DoE (Fig. 3D, see supplementary information for calculation) and several orders of magnitude lower than an exhaustive search to fully screen the design space, assuming a grid of 10 levels (Fig. 3D). We acknowledge the full-screen design is infeasible in most practical settings. Since the BO algorithm considers the design space as a continuum (instead of discretized space used in most statistical DoEs), the full screen provides an estimate of the spread for the complete design space (approximated with a grid of 10 levels here) (see Supplementary Information).”*

Line 136. “With as few as 12...”. Is this using the optimized media? Is this 12 additional experiments after an initial database is generated for the nine new factors or just 12 experiments? How does this compare to something like a Plackett-Burman Design in terms of number of experiments? Are these in one batch or several batches of smaller numbers of experiments? Expand here or refer to methods.

There was no initial database employed. The 12 experiments were run in two iterations/batches of six experiments each for the purpose of optimizing the mixture of the 8 cytokines evaluated. The iterations 0 and 1 labeled in Figure 2D indicate the experiments planned in the initial and the following iteration. This information is now explicitly specified in the revised manuscript [Ln 172-174].

Plackett-Burman Design is typically used for screening and selecting a subset of factors. Here, the cytokines were preselected based on prior knowledge about known influential cytokines for lymphocytic cell populations *in vivo*. For the 8 factors tested here, a standard PB design would require 12 experiments, and a folded PB design would require 24 experiments.

Line 143. For the PBMC media, would you have ended up with a better media if you had added the growth factors to only 1-3 of the base media instead of four? You can comment on this in the discussion.

If only 1-3 out of the four was the best to maintain viability, the algorithm would have picked that as the optimal solution. For cases that might result in invariant responses, however, the algorithm might not use 0 (but assign a small value) since it is optimizing the design space as a continuum. In other words, selecting 1-3 out of four media types is a soft constraint in the current framing of the problem. We agree that selecting 1-3 of the four base media might be relevant for practical reasons such as cost or minimizing the total number of components in the media. For such cases, this can be addressed by adding additional constraints to the current optimization. We have added a general discussion around this point in the revised manuscript.

Revised Text (Pg 16, Ln 432-436):*“The scalability of the methods here, however, also depends on practical considerations in media optimization, such as the cost, solubility, stability/cross-compatibility, and environmental impact of the components. These considerations could be included in the optimization as additional objectives and (or) constraints and addressed through further advances of the current framework.”*

Line 148. It might be good at the first reference of K. phaffii to let readers know that this organism was formerly known as P. pastoris. I think more readers interested in your work will recognize this as an important industrial host.

We have now added this in the revised manuscript [Ln 110-112]

Line 151. “Induce production” I would be specific here and say production of recombinant protein, as that is what you are optimizing here. Production could also mean biomass.

We have now addressed this in the revised manuscript [Ln 186]

Line 152. Reference 42 is not an actual reference. Please add one.

This is an Invitrogen manual and thus had an issue with the referencing software. We have manually corrected this entry.

Line 162. Table 1 is confusing in its current format. Reformat to make the Benchmarks and possible carbohydrate additions more clear. The C additions look like they are already paired up as possibilities which I think is not your intent.

We agree that the table formatting made the carbon additions look paired. We have reformatted Table 1 in the revised manuscript.

Lines 179-181. Discuss what these media are and how different they are from the benchmarks. Are the carbon sources different? How many are chosen or is this constrained?

The optimal media formulation for the three molecules is summarized in Figure 3C. It is constrained to select only one carbon source type per experiment as set up using Co-feed type as a categorical variable. We have now added text to highlight this and included biological insights to add translational value to the optimized media formulations.

Revised Text (Pg 9-10, Ln 223-238): *“Furthermore, the optimal media compositions differed considerably from the current Benchmark media and among the tested molecules with no two molecules converging to the same composition (Fig. 3C), highlighting the unique requirements faced when optimizing media for the efficient production of different recombinant proteins. The alternative carbon sources considered in this work have not been widely studied for their impact on recombinant protein production in K.phaffii. Sorbitol has been used as a co-fed carbon source for generating biomass with K.phaffii⁵¹. L-rhamnose, another carbon source metabolized by K.phaffii⁵², has shown improved production for HSA here, suggesting a potential alternative to sorbitol. Other carbon sources, such as glycerol, glucose, fructose, and mannose, are known to support growth/biomass accumulation while having a repressive impact on the pAOX1 promoter^{51,52}, making them unsuitable candidates to promote protein production. Many carbon sources not known to be metabolized by K.phaffii, including D-arabinose, D-ribose, D-xylose, galactose, lactose, xylitol and sucrose⁵² interestingly showed benefits as a co-feed to enhance recombinant protein production here. Which carbon sources were beneficial, however, depended on the protein produced. How these different carbon sources influence protein production would merit further investigations to assess their influence on the cellular states.”*

Lines 188-191. Not sure where these numbers come from and what a comprehensive search means.

We have reworded the sentence for clarity [Ln 246-247]. By comprehensive search, we meant the Full-Screen method. As noted previously: Bayesian Optimization (in contrast to the statistical DoE approaches), considers the entire continuum of the design space instead of discretizing it. As noted in response to a similar comment above (comment associated to Lines 131-132), an exhaustive search with a finer grid (here approximated with 10 levels) was added as a comparison to demonstrate the spread of the design space [Ln 247-250].

- Please clarify for each of the optimization methods, what is the optimization method and how were the numbers calculated.

We have added a section in the supplementary information (SI Section 1) to describe the calculation of the numbers determined for the different methods.

We have also expanded the number of experiment calculations for a couple of other options of DoE based on the reviewer's overall comments.

Lines 196-198. Did you test other initial designs? It may not make that much difference, but it seems like it could.

We did not test other initial designs in this work. We agree that this parameter is another potential area for exploration. We focused, however, on using space-filling designs, specifically Latin Hypercube Sampling (LHS) when applicable, and random sampling otherwise, as both offer broad coverage of the design space (unlike many traditional DoE methods). This approach has proven successful in Bayesian Optimization (BO) applications, both in our past work and in studies by others. We have added additional references to past work on this aspect.

Revised Text (Pg 10, Ln 255-259): *“First, the use of a space-filling design to generate the initial iteration of experiments maximized the variability in the input-output combinations seen by the GP model, thus, allowing for an efficient initial representation of the system by the model. The use of such space-filling designs to generate initial data has shown success in several applications^{27,28} including media optimization^{31,32}.”*

Furthermore, we have acknowledged that there might be an impact of this choice and the number of experiments needed for convergence in the methods section where the initial design is explained.

We have also added reference to interesting alternatives from recent literature that can be used as potential options to generate initial designs similar in philosophy or are interesting modifications to the space-filling designs in the Discussions section of the revised manuscript.

Revised Text (Pg 16, Ln 423-425): *“Furthermore, the space-filling designs used can be modified to account for constrained design spaces and categorical variables⁵⁵⁻⁵⁹, thus, improving the efficiency of initial design and iterative-sampling.”*

- How about the number of experiments in the initial design?

The number of experiments in the initial design was defined by the throughput of the system and the material availability. For the *K.phaffii* carbon source optimization, 23 initial experiments (+1 control) were used, followed by 11 experiments (+1 control) in each of the future iterations. For the 9-factor design space, however, 11 experiments (+1 control) were used in all the iterations. For the PBMC case study, we used batches of 6 experiments each in all iterations, including the initial round. These details were previously available on the corresponding figures (via color shadings to indicate different iterations). In the revised manuscript, we have added the details explicitly at

the relevant places in the results section as well [Ln 158-159, Ln 172-174, Ln 241-242].

- Will this impact convergence of your algorithm? What is the trade off?
In our case, the initial number of experiments and the batch size were limited by the experimental setup and material. If not, we believe that both these parameters could have an impact on convergence. While the initial design determines the efficiency of the initial GP model built, a poor choice of batch size could impact the total number of experiments or iterations required to reach convergence. We have added this point to the Methods section of the revised manuscript.

Revised Text (Pg 23, Ln 618-623):” *We note here that the choice of batch size may impact convergence, either by increasing the number of iterations required (selecting small batch size) or the total number of experiments required (selecting large batch sizes). We also note that the batch sizes could be modified as the iterative learning progresses instead of having a fixed batch size.*”

Lines 208-234. These results seem to match what others have found. I’m not sure how novel this is, but it is definitely interesting. Could each of these figures be placed in the sections with their respective optimizations? This might make more sense than going back and forth between case studies.

We agree that the suggested approach is an alternative way of organizing the data and figures and we definitely considered this structure while drafting the manuscript! We believe the organization provides support that the approach works in two examples, and then allows us to examine more closely the similarities and differences in the two applications.

Lines 244-248. Can you assure that the algorithm found the best combination or just a good combination? It didn't exhaustively search the space. What if you went further? Where is the trade-off? You could comment on this in the discussion. You seem to be basing your end point on the model predictions matching the experiment. This makes sense, but how do you know that your model doesn't match experiment in an unexplored region that happens to have a better optimum?

We believe that we did find the optimal target value and would reach the same optimal target values if we went further. However, given the complex non-linear response surface, proceeding further will likely plan experiments to find more combinations yielding similar target values. We have added this point to the discussion around thus point in the results section.

Revised Text (Pg 8-9, Ln 208-214): *“These identified media compositions correspond to the best achievable targets for the considered optimization task. In this case, for each molecule, we identified multiple and distinct compositions of media (3 compositions for RBDJ, 2 for HSA, and 2 for trastuzumab) that could optimize protein production to a similar degree (Fig. 3C). We note, however, that given the complex non-linear response surface, using more experiments and time, further iterative learning cycle could yield additional combinations (of the continuous design factors) of media giving similar target values.”*

Line 260. “to the first case.” Do you mean second case?

No, we meant the first case. In the context of the previous sentence, this implied the transfer learning case for modified design spaces (as opposed to transfer learning to alternate biological systems). We have reworded the statement to clarify this meaning.

Revised Text (Pg 13, Ln 332-333): *“In this work, we explored the feasibility of extending our current framework to the first case, that is, to transfer learning for modified design spaces.”*

Lines 268-274. This seems like a unique idea that could be very helpful as cases change for optimization. Do you think you will reach the same optimum?

We appreciate this comment and agree it is a unique feature. We believe that we would reach the same optimal target values but given the complex non-linear response, there could likely be other combinations yielding similar target values. In this particular type of transfer learning, the prior data just probes a subspace of the design space (Spanning 4 out of the 9 factors with the new 5 factors essential at a fixed value). The design of the acquisition function results in experiments being planned at the higher uncertainty regions that correspond these previously unexplored design spaces (from the new 5 factors). This idea was validated for the experiments planned (Fig. S6). Having said that, if the optimization was already performed considering all the 9 factors, a minimum of 90 experiments would be required, if not more. In contrast, by using the existing data, this load could be reduced to 72 experiments (20% reduction).

Revised Text (Pg 13-14, Ln 356-359): *“This ability to use prior learning in the form of a surrogate model thus resulted in at least a 20% reduction in the experiments that started from scratch using the BO approach for the new design space which would have required at least 90 experiments (based on the carbon source optimization case).”*

Lines 283-285. Your comparison seems unlikely, as there are definitely DoE methods for looking through 12 variables in less than 200 experiments.

We agree that there are DoE methods to implement with a lot fewer experiments, albeit with a trade-off in resolution and implemented with methods like RSM. This is valid, however, only for continuous/quantitative variables. For categorical variables, especially with a large number of categories (e.g., 19 categories in this case) and multiple categories, the lack of connectivity between the different categories makes the optimization problem more complex, requiring a lot more experiments.

We have added calculations for the number of experiments for the considered approaches for both case studies which we believe supports these calculations (SI Section 1). We have also added an estimate of a potential resource-conservative DoE approach to this comparison now based on the reviewer's suggestion (Figure 2C and 3D).

Discussion:

Lines 308-311. Again, efficient DoE methods will take a lot fewer experiments than what you have listed in your figures. While this makes your method look really good compared to DoE, it's unfortunate because people familiar with DoE may discount your work. Your method is likely better than really efficient DoE methods. Because you haven't tried one of these experimentally for comparison, you could estimate differently or refer to previous work that actually did these comparisons.

We agree that there are DoE methods to implement with a lot fewer experiments, albeit with a trade-off in resolution. It may be feasible for a DoE to fit practical needs or budget ('efficient' in terms of the number of experiments compared to some of the standard implementations of DoEs), but these assumptions can result in sub-optimal solutions as highlighted in some of the references suggested by the reviewer and other works. In some cases, even where the DoE methods are implemented "efficiently", the outcomes may not surpass those from other approaches of this type.

We do acknowledge the point that here the comparison is made to give an estimate of the number of experiments. As noted in previous responses, we have now included the following in the revised manuscript:

- 1. Calculations of the number of experiments are added to the Supplementary information (SI Section 1)**
- 2. Added comparison for the number of experiments calculated for additional DoE approaches in-line with the comments raised by the reviewer. (Figure 2C and 3D)**

Lines 316-320. You haven't proven that the transfer learning is better than starting over or larger from beginning. You are simply showing that it works ok.

We would like to clarify that we are not claiming that one shouldn't start over or that transfer learning gives better optimal solutions. The goal here was to highlight that we can accumulate learnings across campaigns using such an approach that can help reduce resources for future campaigns. The expected minimum number of experiments to reach optimal solutions when starting from scratch is at least 90 experiments (based on the sole carbon source optimization case). By using the prior learning in this transfer learning approach, we only needed 72 experiments, thus, resulting in a 20% reduction in experimental requirements. We have clarified this in the revised manuscript [Ln 358-364].

Revised Text (Pg 13-14, Ln 356-359): *“This ability to use prior learning in the form of a surrogate model thus resulted in at least a 20% reduction in the experiments that started from scratch using the BO approach for the new design space, which would have required at least 90 experiments (based on the carbon source optimization case).”*

Lines 331-332. There is active feedback from in DoE from iteration to iteration (and there will almost always be iterations)--just not an accumulation of knowledge, making it less efficient.

We thank the reviewer for highlight this point and have now modified the language to clearly communicate this message [Ln 408-410].

Lines 335-336. This assumes that the ranges don't move in DoE. This would be a bad DoE approach that people wouldn't normally use. The advantage is more that DoE is a gradient optimization method and is more likely to get stuck in local minima than your method.

We implied the following here: For a given campaign, in a DoE, the design space is discretized to consider levels (usually high, low, and sometimes centers or axial points). This is in contrast to the proposed approach, which considers the entire continuum and covers the design space. Gradient-based optimization is used after such screening designs via either successive DoE or surrogate modeling such as RSM, ANN, metaheuristics, etc. The local minima encountered by gradient optimization in these approaches are likely due to the insufficient capturing of interactions using the discretized design spaces considered by DoEs. We have clarified this in the revised manuscript.

Revised Text (Pg 15, Ln 413-415): *“Finally, the approach here offers broader coverage of a design space compared to DoE studies that discretize the design space and only test the corner and center points of the design space in a defined campaign.”*

Methods:

Lines 376-380. These two sentences are likely copied from a protocol. Reword for the paper.

We have reworded this description in the revised manuscript to enhance readability and avoid the artifact pointed out by the reviewer [Ln 466-470].

Lines 380-382. Clarify how you combined replicates. This sentence is not clear and not consistent with subsequent phrasing later in the manuscript on Lines 385-386.

We have reworded this description in the revised manuscript to enhance clarity [Line 470-476].

Lines 410-412. Reword to make sense.

We have rephrased this part in the revised manuscript for clarity [Line 502-508].

Line 414. Change “outgrown” to “grown.”

This is now implemented in the revised manuscript.

Lines 420-421. Explain how duplicates give enough data to describe variability. Also, it is not clear how you use control experiments across batches to assess variability. Please clarify.

We have now explained the use of control experiments and the subsequent process noise calculation and incorporation in more detail in the revised manuscript.

Revised text (Pg 19, Ln 514-521): *“Biological duplicates were run for all designed experiments and the average value of the biological duplicate was considered for the modeling. Cultivation using media indicated in Benchmark 1 was run as a control experiment on each iteration. The process noise was calculated as the variance across the control experiment for all instances including the contemporaneous iteration. For building the initial model, the variance was calculated from a preliminary experimental campaign to select molecules expressed with Benchmark 1 media. This estimated process noise was incorporated into the modeling (see Gaussian Processes).”*

Revised text (Pg 21, Ln 566-568): *“To this kernel, process noise was added through a white noise kernel with a fixed variance computed based on the variation in the replicates of the control experiment for each iteration/across different iterations.”*

Lines 425-426. This is not a sentence. Please edit.

We have corrected this in the revised manuscript.

Figures and Tables:

Lines 654-657. What is iteration 0 and iteration 1 in Figure 2d? Figure 2c numbers need to be corrected as per note above.

Iterations 0 and 1 in Figure 2D are the initial and first rounds of cytokine optimization. We have added a sentence in the main text of the revised manuscript to clearly communicate this [Ln 172-174]. The calculation for the number of experiments in Figure 2C is provided in the supplementary information (Table S1) of the revised manuscript. We have also augmented Figure 2C to include the number of experiment calculations for additional DoE approaches.

Figure 3c. The rows of this chart are not clear. Please modify. Again figure 3c numbers seem to be off—or at least need better explanation.

The calculation for the number of experiments in Figure 3C is provided in the supplementary information of the revised manuscript (Table S2). We have also augmented Figure 3C to include the number of experiment calculations for additional DoE approaches.

Figure 4b is difficult to interpret. Could this be shown in one 3D graph? Figure 4c would be easier to interpret with different colors or patterns, as two of the colors look identical in gray scale.

We appreciate the suggestion of the reviewer. However, since there are 4 variables, it won't be possible to consolidate the data into a single 3D graph. We believe that having multiple 3D graphs will make interpretability further complicated. We have, however, now modified the text describing the figure to enhance interpretability. Additionally, in line with the comment for Figure 5 below, we have added a univariate plot of the evolution of different design factor values over time in the Supplementary information (Fig. S4) and appropriately referenced it in the main text.

Revised Text (Pg 11, Ln 276-283): *“For the example of media blending for PBMCs, we observed in the pairwise plot of design space (Fig. 4B) as well as the univariate plot of individual design factors over iterations (Fig. S4), that exploration dominated the first two iterations (Iterations 0 and 1), resulting in cell viability varying from 5% to 75% over (Fig. 4A). Subsequently, Iteration 2 included a mix of exploration and exploitation: For instance, Blend 15 exploited a previously observed region covered by Blend 12 (Fig. 4B,C; Fig. S4). Subsequently, the final iteration (Iteration 3) exploitatively reduced the search space to a specific ratio of DMEM with a focus on perturbations involving different combinations of the other media types (Fig. 4B, C; Fig. S4).”*

In the revised manuscript, we have modified Figure 4C to include patterns and aid visual interpretation of the figure.

Figure 5. The exploration and exploitation are difficult to see in these figures. Is there a

way to show how the values of the different factors change over time instead?

We thank the reviewer for this comment. We have now added a supporting figure of the univariate evolution of different design factors over the iterations in the Supplementary information (Fig. S5) to aid in the interpretation of current Figure 5 and modified the main text to reflect this.

Revised Text (Pg 11-12, Ln 287-299): *“Unsurprisingly, the algorithm planned experiments in Iteration 1 for types of carbon sources not probed in the initial experiments, as observed in the pairwise plot of the design space (Fig. 5B) and the univariate plot of the individual design factors (Fig. S5) over the different iterations. This outcome can be attributed to the objective function being dominated by the higher uncertainty manifested in these regions of the design space. Iterations 3 through 6, however, emphasized exploitation as observed by the higher fraction of experiments planned in a limited region of the design space, resulting in increased specific productivity (Fig. 5A) with limited testing of other regions of the design space (Fig. 5B). Particularly, the favorable part of the design space corresponding to the continuous variables, Glycerol and Methanol, were identified in Iterations 3 and 4 (Fig. S5A, B). However, for the categorical variable (co-feed type) and the categorical-coupled continuous variable (Co-feed concentration), Iterations 3 and 4 focused on navigating different carbon sources at a limited concentration range, while iterations 5 and 6 limit the co-feed type to the most favorable ones, exploring a range of concentration for these (Fig. S5C, D).”*

References for Review:

1. Claes, E., Heck, T., Coddens, K., Sonnaert, M., Schrooten, J., & Verwaeren, J. (2024). Bayesian cell therapy process optimization. *Biotechnology and Bioengineering*, 121, 1569–1582. <https://doi.org/10.1002/bit.28669>
2. Coleman, M.C. and Block, D.E. (2007), Nonlinear experimental design using Bayesian regularized neural networks. *AIChE J.*, 53: 1496-1509. <https://doi.org/10.1002/aic.11175>
3. Cosenza Z, Block DE, Baar K, Chen X. Multi-objective Bayesian algorithm automatically discovers low-cost high-growth serum-free media for cellular agriculture application. *Eng Life Sci.* 2023; 23:e2300005. <https://doi.org/10.1002/elsc.202300005>
4. Cosenza, Z., Astudillo, R., Frazier, P. I., Baar, K., & Block, D. E. (2022). Multi-information source Bayesian optimization of culture media for cellular agriculture. *Biotechnology and Bioengineering*, 119, 2447–2458. <https://doi.org/10.1002/bit.28132>
5. K. Watanabe, T.-Y. Chiou, M. Konishi, Optimization of medium components for

protein production by *Escherichia coli* with a high-throughput pipeline that uses a deep neural network, *Journal of Bioscience and Bioengineering*, Volume 137, Issue 4, 2024, Pages 304-312, ISSN 1389-1723 <https://doi.org/10.1016/j.jbiosc.2024.01.005>.

6. Xiao, W., Shi, X., Huang, H., Wang, X., Liang, W., Xu, J., Liu, F., Zhang, X., Xu, G., Shi, J., & Xu, Z. (2024). Enhanced synthesis of S-adenosyl-L-methionine through combinatorial metabolic engineering and Bayesian optimization in *Saccharomyces cerevisiae*. *Biotechnology Journal*, 19, e2300650. <https://doi.org/10.1002/biot.202300650>

7. Zhang, G. and Block, D.E. (2009), Using highly efficient nonlinear experimental design methods for optimization of *Lactococcus lactis* fermentation in chemically defined media. *Biotechnol Progress*, 25: 1587-1597. <https://doi.org/10.1002/btpr.277>

8. Zhang G, Mills DA, Block DE. Development of chemically defined media supporting high-cell-density growth of lactococci, enterococci, and streptococci. *Appl Environ Microbiol*. 2009 Feb;75(4):1080-7. doi: 10.1128/AEM.01416-08. Epub 2008 Dec 12. PMID: 19074601; PMCID: PMC2643557.

Point-By-Point Response

Reviewer #1 (Remarks to the Author):

I find the response to my comments largely satisfactory, both in the response letter as well as in terms of corresponding modifications of the paper.

We thank the reviewer for the comments and feedback that helped us strengthen the manuscript.

However, I find it very peculiar that they do not respond in any way to my comment on a recently published work proposing a very similar methodology to essentially the same application, that is, optimizing cell culture media composition using a Bayesian like approach (Wang et al., 2024, full reference in my original review). Although Wang et al. only demonstrated the method on a simulation model, it is highly relevant to the current work and should hence be commented upon and referenced.

As this point was not included among the detailed comments or questions in the previous revision, we may have inadvertently missed providing a direct response to it.

We would like to highlight that we had already included references to recent works that employ a similar approach for media optimization, involving actual experimental validation.

Some studies have also demonstrated these approaches for designing and optimizing cell culture media considering multiple objectives³³ and information sources³⁴.

In response to the current comments, we have also included the reference suggested by the reviewer in the revised manuscript in this statement.

Revised Text: Some studies have also demonstrated these approaches for designing and optimizing cell culture media³³⁻³⁵ considering multiple objectives³³ and information sources³⁴. These, however, use only continuous design factors in their optimizations

Reviewer #2 (Remarks to the Author):

All my concerns have been addressed, and the revised manuscript is improved a lot. I recommend this work to be published!

We thank the reviewer for the comments and feedback that helped us strengthen the manuscript.